# TimePro: Efficient Multivariate Long-term Time Series Forecasting with Variable- and Time-Aware Hyper-state

**Xiaowen Ma** [* 1]  **Zhenliang Ni** [* 1]  **Shuai Xiao** [1]  **Xinghao Chen** [1]

**https://github.com/xwmaxwma/TimePro**

## Abstract

In long-term time series forecasting, different variables often influence the target variable over distinct time intervals, a challenge known as the multi-delay issue. Traditional models typically process all variables or time points uniformly, which limits their ability to capture complex variable relationships and obtain non-trivial time representations. To address this issue, we propose TimePro, an innovative Mamba-based model that constructs variate- and time-aware hyper-states. Unlike conventional approaches that merely transfer plain states across variable or time dimensions, TimePro preserves the fine-grained temporal features of each variate token and adaptively selects the focused time points to tune the plain state. The reconstructed hyper-state can perceive both variable relationships and salient temporal information, which helps the model make accurate forecasting. In experiments, TimePro performs competitively on eight real-world long-term forecasting benchmarks with satisfactory linear complexity.

## 1. Introduction

Mamba (Gu & Dao, 2024) has shown significant advantages in time series forecasting (Ahamed & Cheng, 2024), characterized by its linear computational complexity, and efficient long-term dependency capture capability. Unlike Transformers, which have quadratic complexity concerning sequence length, Mamba achieves linear complexity, making it highly scalable for long sequences. Mamba can also capture long-term dependencies, which is particularly beneficial in time series forecasting. Furthermore, it can not only handle long series of data but also perform well

---

* Equal contributions. [1]Huawei Noah's Ark Lab. Correspondence to: Xinghao Chen <xinghao.chen@huawei.com>.

*Proceedings of the 42$^{nd}$ International Conference on Machine Learning*, Vancouver, Canada. PMLR 267, 2025. Copyright 2025 by the author(s).

in multivariate modeling (Liang et al., 2024; Ahamed & Cheng, 2024; Ma et al., 2024).

Bi-Mamba+ (Liang et al., 2024) introduces a bidirectional structure and splits time series into smaller segments to more comprehensively model the data, incorporating a forgetting gate to selectively integrate new features with historical ones, thereby retaining historical information over longer ranges. S-Mamba (Wang et al., 2025) marks the time points of each variable through the autonomous linear layer, and extracts the correlation between variables through the bidirectional Mamba layer. It also incorporates a feedforward neural network (FNN) to learn time dependence. TimeMachine (Ahamed & Cheng, 2024) combines four Mamba modules to capture both channel-mixed and channel-independent contexts through multi-scale contextual cues. TSMamba (Ma et al., 2024) further optimizes time series forecasting performance through channel-compressed attention modules and a two-stage training strategy, with experiments showing that its innovative components enhance forecasting accuracy while reducing computational overhead. The mentioned methods often employ different ways to scan features from various directions. However, these methods overlook the fact that different variables have different impact durations on the target variable, i.e., the multi-delay issue, which limits their performance.

The multi-delay issue in multivariate time series forecasting (Xu et al., 2016; Chandereng & Gitter, 2020) is defined as the temporal discrepancy in the propagation of influence from different predictor variables to the target variable, characterized by the presence of distinct and non-uniform time lags between the changes in predictor variables and their corresponding effects on the target variable. However, existing mamba models process all variables or time points in a uniform manner, making it difficult to capture critical time points and obtain non-trivial time representations. Moreover, this problem also occurs in transformer-based models. For example, PatchTST (Nie et al., 2023) captures global temporal dependencies in a channel-independent manner, while treating them uniformly for all variables. iTransformer (Liu et al., 2024b) focuses on modeling variable relationships, yet uniformly performs a coarse linear projection for different

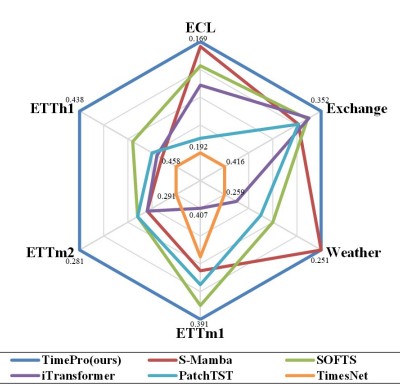

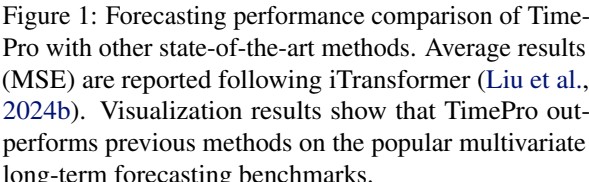

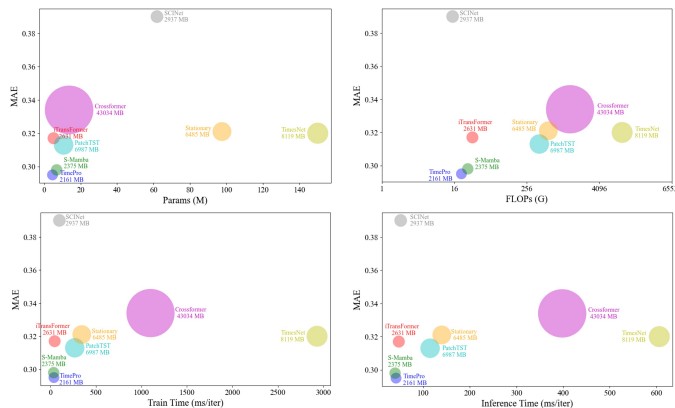

Figure 1: Forecasting performance comparison of Time-Pro with other state-of-the-art methods. Average results (MSE) are reported following iTransformer (Liu et al., 2024b). Visualization results show that TimePro outperforms previous methods on the popular multivariate long-term forecasting benchmarks.

Figure 2: Efficiency comparison of TimePro with other state-of-the-art methods. We set the lookback window L = 96, forecast horizon H = 720, and batch size to 16 in the Electricity dataset. The train and inference times are measured on the Nvidia V100 GPU. Compared to other methods, TimePro achieves satisfactory performance with minimal parameters, FLOPs, memory consumption and competitive training and inference speeds.

time points, making it difficult to capture complex internal temporal variations. To address this limitation, we propose a novel Mamba-based model called TimePro, which incorporates variable and time-aware hyper-state construction. This innovation allows the model to dynamically adapt to the distinct temporal characteristics of each variable, thereby enhancing its ability to capture the complex dynamics of multivariate time series data and effectively deal with the multi-delay issue.

Unlike traditional approaches that merely transfer plain states across variables, TimePro preserves the fine-grained temporal features of each variate token and adaptively selects the focused time points to tune the plain state. Specifically, we first scan the variable dimension to obtain the hidden state, which contains the correlation between variables. This hidden state serves as a foundation for subsequent temporal refinement. Following this, a specialized network is employed to learn the offsets of critical time points. By adaptively selecting these key time points, TimePro dynamically updates the hidden states to reflect the most salient temporal information. This adaptive mechanism enables the reconstructed hyper-state to integrate both variable-specific information and subtle temporal changes, empowering the model to produce highly accurate forecasts.

As shown in Fig. 1 and Fig. 2, TimePro has consistently demonstrated competitive performance across eight real-world forecasting benchmarks. Notably, it achieves these results while maintaining linear complexity, ensuring that computational efficiency is not compromised. This balance between accuracy and efficiency makes TimePro a powerful tool for tackling complex multivariate time series forecast-

ing tasks, particularly those involving long sequences or high-dimensional data. Our main contributions are as follows:

- We devise a novel time-tune strategy that adjusts the variable states by adaptively selecting important time points and uses reconstructed hyper-states to obtain the output. The ability of the hyper-state to perceive complex variable relationships and intra-variable time changes facilitates accurate prediction.

- By combining the hyper-state reconstruction and hardware-aware implementation, we propose an efficient multivariate long-term time series prediction model TimePro.

- TimePro achieves competitive performance on eight real-world datasets, significantly surpassing existing mamba and transformer-based methods.

## 2. Related Work

### 2.1. Time Series Forecasting

In recent years, various methods have emerged in the field of time series prediction (Nie et al., 2024a;b), including those based on RNNs, MLPs, Transformers, and Mamba. RNN-based methods (Hewamalage et al., 2021; Rangapuram et al., 2018; Hou et al., 2025), once dominant, have gradually been overshadowed by newer approaches due to their limited capacity to model long-range dependencies and their high computational complexity, which often leads

to inefficient training and inference. The MLP-based models include DLinear (Zeng et al., 2023), LightTS (Zhang et al., 2022) and TimeMixer (Liu et al., 2024a). The MLP model has a small number of parameters and low computational complexity, but the ability to model long sequence feature relationships is weak. The Transform-based method has strong global information modeling ability and is helpful to capture long-term dependencies in time series (Hou et al., 2024). For example, iTransformer (Liu et al., 2024b) and PatchTST (Nie et al., 2023). However, the transformer method has high computational complexity and is prone to overfitting in time series forecasting tasks. Mamba-based methods can model long time series relationships and have linear complexity, so this kind of method has great advantages in time series prediction tasks (Wang et al., 2025; Ahamed & Cheng, 2024; Liang et al., 2024).

### 2.2. Mamba

In recent years, a series of mamba-based methods have been proposed for time series forecasting. Bi-Mamba+ (Liang et al., 2024) introduces a bidirectional structure and splits time series into smaller segments to more comprehensively model the data, incorporating a forgetting gate to selectively integrate new features with historical ones, thereby retaining historical information over longer ranges. S-Mamba (Wang et al., 2025) marks the time points of each variable through the autonomous linear layer, and extracts the correlation between variables through the bidirectional Mamba layer. In addition, S-Mamba also incorporates a feedforward neural network (FNN) to learn time dependence. TimeMachine (Ahamed & Cheng, 2024) combines four Mamba modules to capture both channel-mixed and channel-independent contexts through multi-scale contextual cues. TSMamba (Ma et al., 2024) further optimizes time series forecasting performance through channel-compressed attention modules and a two-stage training strategy, with experiments showing that its innovative components enhance forecasting accuracy while reducing computational overhead. SST (Xu et al., 2024) deals with long-range global patterns and short-range local changes simultaneously by Mamba and LWT, which solves the shortage of traditional models in modeling long-term and short-term features (Liu et al., 2023). However, the above methods ignore the different time lags between multivariate and prediction results, which limits their performance.

## 3. Preliminaries

**State Space Models (S4).** State space models (SSMs) are proposed in deep learning as common sequence models (Gu et al., 2022), which transform an input sequence $x(t) \in \mathbb{R}^{L_s \times D_s}$ into an output sequence $y(t) \in \mathbb{R}^{L_s \times D_s}$ by utilizing a learnable hidden state $h(t) \in \mathbb{R}^{N_s}$. The process could be denoted as follows:

$$\begin{aligned} h'(t) &= \boldsymbol{A}h(t) + \boldsymbol{B}x(t), \\ y(t) &= \boldsymbol{C}h(t), \end{aligned} \tag{1}$$

where $\boldsymbol{A} \in \mathbb{R}^{N_s \times N_s}$ is the evolution parameter, $\boldsymbol{B}, \boldsymbol{C} \in \mathbb{R}^{N_s \times D_s}$ denote the learnable projection parameters, and $N_s$ is the state size.

**Discretization.** The above continuous-time SSMs are not well compatible with deep learning algorithms. Therefore, discretization is needed to align the model with the sampling frequency of the input signal to improve computational efficiency (Gu et al., 2021). Following the previous work (Gupta et al., 2022), given the sampling time scale parameter $\boldsymbol{\Delta}$, the above continuous SSMs are discretized through zero-order hold rule, thus converting the continuous-time parameters $(\boldsymbol{A}, \boldsymbol{B})$ to their corresponding discrete counterparts $(\overline{\boldsymbol{A}}, \overline{\boldsymbol{B}})$:

$$\begin{aligned} \overline{\boldsymbol{A}} &= e^{\boldsymbol{\Delta A}}, \\ \overline{\boldsymbol{B}} &= (\boldsymbol{\Delta A})^{-1}(e^{\boldsymbol{\Delta A}} - \boldsymbol{I}) \cdot \boldsymbol{\Delta B}. \end{aligned} \tag{2}$$

Then, The discretized formulation of Eq. 1 is formulated as:

$$\begin{aligned} h_t &= \overline{\boldsymbol{A}}h_{t-1} + \overline{\boldsymbol{B}}x_t, \\ y_t &= \boldsymbol{C}h_t, \end{aligned} \tag{3}$$

where $\overline{\boldsymbol{A}} \in \mathbb{R}^{N_s \times N_s}$, $\overline{\boldsymbol{B}} \in \mathbb{R}^{N_s \times D_s}$. In addition, the iterative process described in Eq. 3 can be performed by the parallel computing mode of global convolution (Gu & Dao, 2024) to improve the computational efficiency:

$$\begin{aligned} y &= x \circledast \overline{\boldsymbol{K}}, \\ \text{with} \quad \overline{\boldsymbol{K}} &= (\boldsymbol{C}\overline{\boldsymbol{B}}, \boldsymbol{C}\overline{\boldsymbol{A}}\overline{\boldsymbol{B}}, \cdots, \boldsymbol{C}\overline{\boldsymbol{A}}^{L-1}\overline{\boldsymbol{B}}), \end{aligned} \tag{4}$$

where $\circledast$ denotes the convolution operation, and $\overline{\boldsymbol{K}} \in \mathbb{R}^L$ serves as the kernel of the SSMs.

**Selective State Space Models (S6).** Conventional SSMs (i.e., S4) have been implemented to capture sequence context under linear time complexity, despite the fact that they are constrained by static parameterization and cannot perform content-based reasoning. To address this problem, the selective state space model (i.e., Mamba (Gu & Dao, 2024)) has been proposed, which allows the model to selectively propagate or forget information based on the latitude of the current token along the length of the sequence by simply setting the parameters of the SSM as a function of the inputs. In S6, the parameters $B$, $C$, and $\Delta$ are computed directly from the input sequence $x(t)$, thus enabling sequence-aware parameterization.

## 4. Method

Multivariate Time Series Forecasting (MTSF) deals with time series data containing multiple variables or channels at

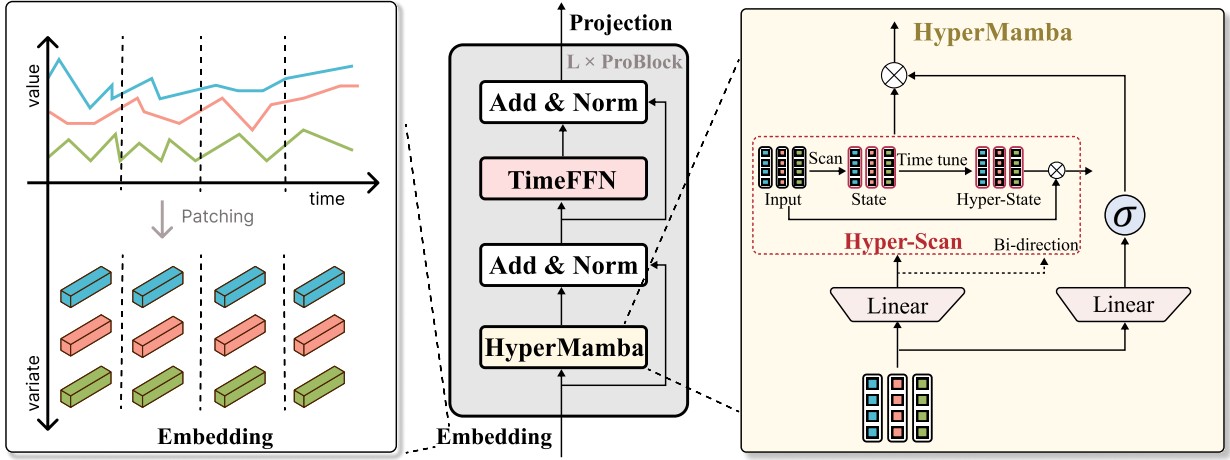

Figure 3: Overview of our TimePro method. The multivariate time series is first embedded along the temporal dimension with the patching operation to get the series representation for each variable. Then the variable correlation and time representation of variables are captured by multiple layers of ProBlock modules. The core component of Problok is HyperMamba, which adaptively selects important time points to regulate the plain state of the variable dimension. The reconstructed time- and variable-aware hyper-states are then applied to obtain the output.

each time step. Given a historical value $\boldsymbol{X} \in \mathbb{R}^{N \times L}$, where $L$ denotes the length of the lookback window and $N$ denotes the number of variables, the goal of MTSF is to predict a future value $\boldsymbol{Y} \in \mathbb{R}^{N \times H}$, where $H > 0$ is the forecast horizon. In this paper, we focus on long-term forecasting $H >= 96$, which is more challenging.

## 4.1. Overview

As shown in Fig. 3, TimePro adopts a transformer-like pure encoder architecture (Vaswani et al., 2023), which consists of the following components:

**Reversible instance normalization** Training and test data often suffer from distributional shifts, and directly inputting raw sequences into the model is not conducive to stable predictions. Following previous work (Liu et al., 2024b; Han et al., 2024), we use reversible instance normalization (RevIN) (Kim et al., 2022) to concentrate the sequence to zero mean and scale to unit variance before inputting the original sequence $\boldsymbol{X}$ into the model. Then, the predicted sequence is reverse normalized to obtain the output $\boldsymbol{Y}$.

**Time and variable preserved embedding** Previous works have either used sequential embedding (Liu et al., 2024b; Wang et al., 2025; Han et al., 2024) or channel-independent patch embedding (Nie et al., 2023). However, this style of uniform processing of all variables or time points limits the ability of models to capture complex variable relationships and obtain non-trivial time representations. Therefore, we use time and variable preserved embeddings, which facilitate the construction of subsequent hyper-states. Specifically, we divide each input univariate

time series $\boldsymbol{X}_{i,:} \in \mathbb{R}^L$ into overlapping patches and preserve the dimensions of the variables,

$$\mathcal{E}_0 = \text{Embedding}(\boldsymbol{X}), \tag{5}$$

where embedding $\mathcal{E}_0 \in \mathbb{R}^{N \times P \times D}$, $P$ and $D$ denote the number and feature dimensions of patches, respectively.

**ProBlock** We apply multiple layers of ProBlock to optimize the embedding, which enables information interactions across time and variables,

$$\mathcal{E}_{i+1} = \text{ProBlock}(\mathcal{E}_i), \ i = 0, 1, \ldots, \gamma - 1, \tag{6}$$

where $\gamma$ is the number of layers. In addition, as shown in Fig. 3, ProBlock is composed of HyperMamba and TimeFFN, which complete the information interaction between variables and the capture of complex changes within variables respectively. We'll describe the HyperMamba module in the next subsection, which is the focus of this paper.

**Linear projection** After $\gamma$ layers of ProBlock, we first flatten the embedding of each variable, and then apply a simple linear projection to obtain the forecast results,

$$\boldsymbol{Y} = \text{Projection}(\mathcal{E}_L). \tag{7}$$

## 4.2. HyperMamba Module

We propose HyperMamba to optimize the modeling of variable dependencies. Existing popular methods (Liu et al., 2024b) mainly use attention to capture this relationship. However, they suffer from quadratic complexity with the number of variables, which severely limits the deployment

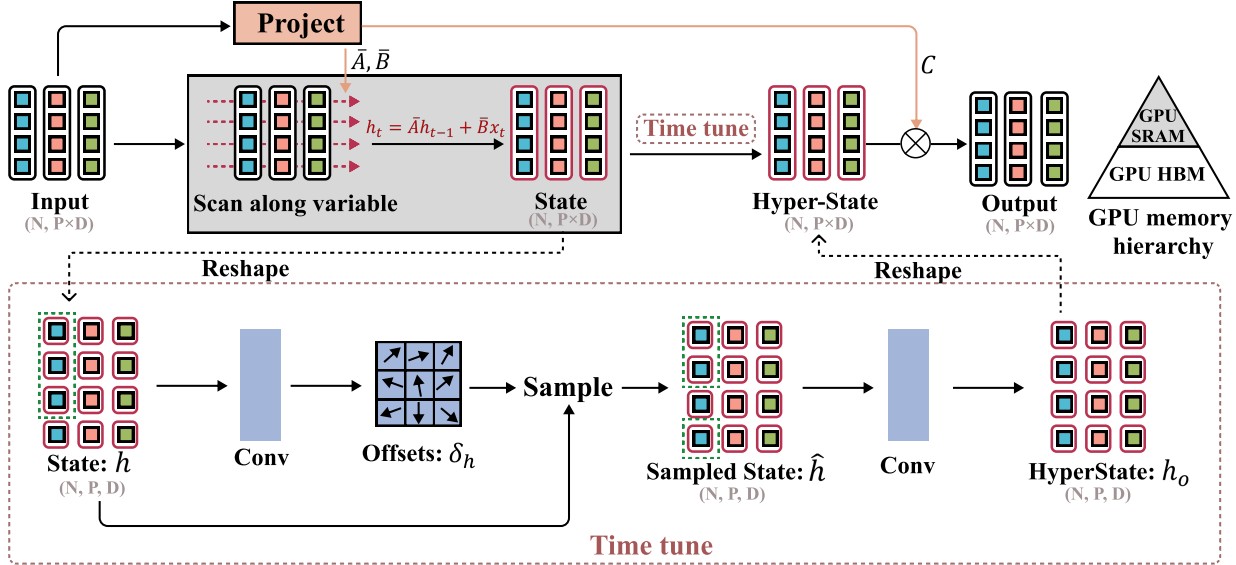

Figure 4: Implementation details of hardware-aware hyper-scan. We effectively apply the GPU memory hierarchy, i.e., perform plain state acquisition on GPU SRAM (implemented in the grey box above), and other operations on GPU HBM. Specifically, we follow the original Mamba implementation by first scanning the embedding along the variables and acquiring the plain state. Then, we perform a reshape on the plain state to recover the fine-grained time dimension of the embedding. Next, we adaptively select important time points for each variable to adjust the plain state to obtain time- and variable-aware hyper-states. Finally, the reconstructed hyper-states are applied to obtain the augmented embeddings through a gating mechanism.

of models in real-world scenarios. Although some methods introduce Mamba (Wang et al., 2025) or MLP (Han et al., 2024) to extract variable dependencies, they use sequence embedding to process each variable sequence, which is not enough to capture complex internal time changes within variables. In particular, in the real world, the relationship between various physical variables is not static and tends to fluctuate with changes in local time (Zhou et al., 2021). Therefore, we propose a novel time tune strategy to construct the variable- and time-aware hyper-state, so as to model the information interaction between variables better.

Specifically, for input $\mathcal{E} \in \mathbb{R}^{N \times (P \times D)}$, we first get $\mathcal{E}_t \in \mathbb{R}^{N \times (P \times D)}$ and $\mathcal{E}_z \in \mathbb{R}^{N \times (P \times D)}$ through two linear projections. Then, we split the $\mathcal{E}_t$ into two parts along the channel and input the Hyper-scan module along the opposite variable direction to capture the global variable dependencies,

$$\hat{\mathcal{E}}_{t1} = \text{Hyperscan}_{1 \to N}(\mathcal{E}_t[:, 0 : \frac{PD}{2}]),$$

$$\hat{\mathcal{E}}_{t2} = \text{Hyperscan}_{N \to 1}(\mathcal{E}_t[:, \frac{PD}{2} : PD]), \qquad (8)$$

$$\hat{\mathcal{E}}_t = \text{Concact}(\hat{\mathcal{E}}_{t1}, \hat{\mathcal{E}}_{t2}).$$

Finally, we apply the SiLU activation function (Ramachandran et al., 2017) to $\mathcal{E}_z$ and dot-multiply it with the augmented $\hat{\mathcal{E}}_t$ to get the output $\hat{\mathcal{E}}$,

$$\hat{\mathcal{E}} = \hat{\mathcal{E}}_t \cdot \text{SiLU}(\mathcal{E}_z). \qquad (9)$$

**HyperMamba Design** The structure diagram of the HyperMamba module can be seen in Fig. 3, which is slightly modified from the original Mamba (Gu & Dao, 2024). First, we replace the selective scan with a hardware-aware Hyper Scan that underpins the construction of hyperstates. Compared to the original selective scan module, the hyper scan does not reduce the efficiency and achieves better performance. In addition, we remove the depthwise convolution before the scan and the linear projection after the scan, which are verified as unnecessary in Sec. 5.3. In addition, we add the scanning inception design, i.e., we split two parts along the channel and scan in opposite variable directions, thereby enhancing the model's ability to capture global variable dependencies.

**Hyper-scan** The Vanilla Mamba transfers the state independently along each dimension of the embedding and gets the output through the parameter $C$. Therefore, when we apply the original selective scanning module to scan along the variable dimension, the state contains only variable information. It is not conducive to capturing accurate variable dependencies, which fluctuate with time changes within the variable. Therefore, we introduce a time tune strategy to adjust the initial state adaptively according to the time points intra-variable.

The process of Hyper-scan is shown in Fig. 4. For the input $\mathcal{E}_t \in \mathbb{R}^{N \times (P \times D)}$, it is first scanned along the variable

dimension and the initial state $h \in \mathbb{R}^{N \times (P \times D)}$ is obtained. This step is implemented in SRAM to reduce the repeated read and write of HBM memory. We then reshape the dimensions of state $h$ and obtain an initial shift $\delta_h$ through a convolution,

$$\delta_h = \text{Conv}(h). \tag{10}$$

Next, we add the reference point to the learnable offset to get the sample point, which serves as the final coordinate for extracting the important time point from the state. In practice, we follow (Xiong et al., 2024) by linear interpolation $\psi$ to make the sampling process differentiable,

$$h_{samp} = h_{ref} + \delta_h, \tag{11}$$

$$\hat{h} = h_{ref} + \psi(h; h_{samp}), \tag{12}$$

where $\hat{h} \in \mathbb{R}^{N \times P \times D \times M}$ and $M$ is the number of sampling time points. Then, a linear mapping is used to fuse the sampled time points to obtain the hyperstate $h_o \in \mathbb{R}^{N \times P \times D}$,

$$h_o = \text{Linear}(\hat{h}). \tag{13}$$

Finally, the reconstructed hyper-state is multiplied with the parameter matrix $C$ in Mamba to get the output.

### 4.3. Complexity Analysis

We analyze the complexity of the core component Hyper-Mamba from patch number $P$, feature dimension $D$, variable number $N$, and sampling point $M$. First, the input passes through two linear projections with a complexity of $O(NP^2D^2)$. Then, we get the parameter matrix $A, B, C$ by linear mapping and structured operation respectively, which has a complexity of $O(NPD)$ since we set the number of states in each dimension to 1. During the scanning process, the initial state is $O(NPD)$ complex because only needs to be scanned independently along the variable dimension. The time tune process involves the generation of offsets (i.e., convolution) and linear projection, requires a complexity of $O(NPMD)$. Given that $D$ is a relatively small constant and M is also a constant (i.e., 9), these two terms can be ignored. In addition, $P$ is affected by the length of the series $L$, so the complexity of the entire hyperMamba is $O(NL)$. As shown in Table 1, two classical comparison models, such as iTransformer and PatchTST, require significantly more complexity than TimePro.

Table 1: Complexity comparison between popular time series forecasters concerning series length $L$, number of variables $N$. TimePro achieves only linear complexity.

| | TimePro (ours) | iTransformer | PatchTST | Transformer |
|---|---|---|---|---|
| Complexity | $O(NL)$ | $O(N^2 + NL)$ | $O(NL^2)$ | $O(NL + L^2)$ |

## 5. Experiments

### 5.1. Datasets and Implementation Details

In order to comprehensively evaluate the performance of the proposed TimePro, we conduct extensive experiments on five widely used real datasets, including ETT (4 subsets), Exchange, Electricity, Weather (Zhou et al., 2021; Wu et al., 2021), and Solar-Energy (Lai et al., 2018). All experiments are implemented on four Tesla V100 GPUs and follow the common training settings in (Liu et al., 2024b). A detailed description can be found in Appendix A.

### 5.2. Main Results

As shown in Table 2, TimePro achieved state-of-the-art performance on multiple datasets, with the first place marked in red and the second place marked in blue. TimePro scored 12 firsts and 2 seconds out of 16 metrics. With the exception of the Solar-Energy dataset, our approach outperforms the state-of-the-art methods SOFTS and iTransformer. On the weather dataset, the MSE of our method is 2.0% lower than that of SOFTS and 3% lower than that of iTransformer. On the weather dataset, the MSE of our method is reduced by 2% compared with that of SOFTS and 3.1% compared with that of iTransformer. Compared with SOTFS and iTransformer on the ETTm1 dataset, TimePro reduces mae by 1.2% and 2.9%, respectively. On most data sets, TimePro outperforms past advanced methods such as TimesNet, PatchTST and Dlinear. This is because these methods deal with the time dimension in a unified manner, and there is no way to capture complex time relationships.

Fig. 2 also provides a detailed comparison of efficiency metrics including parameters, FLOPs, training time and inference time. It can be observed that TimePro possesses better computational efficiency than previous methods. Specifically, TimePro has a training time and inference time similar to S-Mamba and significantly outperforms other convolutional or transformer-based methods. For example, TimePro obtains lower prediction errors with 2.7 and 14.4 times the inference speed of PatchTST and TimesNet, respectively. Moreover, TimePro has the minimal parameters, FLOPs, and memory consumption. For example, TimePro requires only 67% parameters and 78% GFLOPs of S-Mamba. In addition to satisfactory efficiency, TimePro also outperforms recent advanced models including iTransformer and S-Mamba in forecasting performance. These results demonstrate TimePro's lightweight and suitability for deployment in a variety of real-world scenarios where resources are constrained.

To further validate the robustness of our approach, we conduct a comparative analysis of various models across different lookback window lengths. As depicted in Figure 7, TimePro consistently achieves the lowest MAE across all

Table 2: Multivariate long-term forecasting results with horizon $H \in \{96, 192, 336, 720\}$ and fixed lookback window length $L = 96$. Results are averaged from all prediction horizons.

| Models | TimePro ours | | S-Mamba (2025) | | SOFTS (2024) | | iTransformer (2024b) | | PatchTST (2023) | | Crossformer (2023) | | TiDE (2023) | | TimesNet (2023) | | DLinear (2023) | | SCINet (2022) | | Stationary (2022) | |
|---|---|---|---|---|---|---|---|---|---|---|---|---|---|---|---|---|---|---|---|---|---|---|
| Metric | MSE | MAE | MSE | MAE | MSE | MAE | MSE | MAE | MSE | MAE | MSE | MAE | MSE | MAE | MSE | MAE | MSE | MAE | MSE | MAE | MSE | MAE |
| ECL | **0.169** | **0.262** | 0.170 | 0.265 | 0.174 | 0.264 | 0.178 | 0.270 | 0.189 | 0.276 | 0.244 | 0.334 | 0.251 | 0.344 | 0.192 | 0.295 | 0.212 | 0.300 | 0.268 | 0.365 | 0.193 | 0.296 |
| Exchange | **0.352** | **0.399** | 0.367 | 0.408 | 0.361 | 0.402 | 0.360 | 0.403 | 0.367 | 0.404 | 0.940 | 0.707 | 0.370 | 0.413 | 0.416 | 0.443 | 0.354 | 0.414 | 0.750 | 0.626 | 0.461 | 0.454 |
| Weather | **0.251** | **0.276** | 0.251 | 0.276 | 0.255 | 0.278 | 0.258 | 0.278 | 0.256 | 0.279 | 0.259 | 0.315 | 0.271 | 0.320 | 0.259 | 0.287 | 0.265 | 0.317 | 0.292 | 0.363 | 0.288 | 0.314 |
| Solar-Energy | 0.232 | 0.266 | 0.240 | 0.273 | **0.229** | **0.256** | 0.233 | 0.262 | 0.236 | 0.266 | 0.641 | 0.639 | 0.347 | 0.417 | 0.301 | 0.319 | 0.330 | 0.401 | 0.282 | 0.375 | 0.261 | 0.381 |
| ETTm1 | **0.391** | **0.400** | 0.398 | 0.405 | 0.393 | 0.403 | 0.407 | 0.410 | 0.396 | 0.406 | 0.513 | 0.496 | 0.419 | 0.419 | 0.400 | 0.406 | 0.403 | 0.407 | 0.485 | 0.481 | 0.481 | 0.456 |
| ETTm2 | **0.281** | **0.326** | 0.288 | 0.332 | 0.287 | 0.330 | 0.288 | 0.332 | 0.287 | 0.330 | 0.757 | 0.610 | 0.358 | 0.404 | 0.291 | 0.333 | 0.350 | 0.401 | 0.571 | 0.537 | 0.306 | 0.347 |
| ETTh1 | **0.438** | **0.438** | 0.455 | 0.450 | 0.449 | 0.442 | 0.454 | 0.447 | 0.453 | 0.446 | 0.529 | 0.522 | 0.541 | 0.507 | 0.458 | 0.450 | 0.456 | 0.452 | 0.747 | 0.647 | 0.570 | 0.537 |
| ETTh2 | 0.377 | 0.403 | 0.381 | 0.405 | **0.373** | **0.400** | 0.383 | 0.407 | 0.385 | 0.410 | 0.942 | 0.684 | 0.611 | 0.550 | 0.414 | 0.427 | 0.559 | 0.515 | 0.954 | 0.723 | 0.526 | 0.516 |

tested lookback windows. Specifically, when the lookback window is set to 48, our method demonstrates a significantly lower MAE compared to SOFTS. This is attributed to TimePro's capability to effectively capture crucial temporal information, enabling it to maintain strong performance even with limited data. Notably, on the ETTm2 and Exchange datasets, we observe a substantial reduction in MAE when the lookback window is extended to 336. While it is generally expected that a larger lookback window would provide more information and thus lead to more consistent model performance, our model's ability to excel under such conditions is noteworthy. This is largely due to TimePro's dynamic feature selection mechanism, which allows it to identify and focus on key temporal features, thereby preventing the relevant information from being overwhelmed by the sheer volume of time series data.

## 5.3. Ablation study

We conduct a series of ablation experiments to investigate the effect of different hyperparameters and the effectiveness of the model structure design.

**Effect on multi-delay issue** We first add a visualization experiment to further validate the validity and interpretability of TimePro for the multi-delay issue, as shown in Fig. 5. We choose test sequences from the ETTm1 and ETTh1 datasets as examples. Specially, we first calculate the correlation of label sequences (i.e., groundtruth) by Pearson correlation coefficient,

$$r_{xy} = \frac{\sum_{i=1}^{L} (x_i - \bar{x})(y_i - \bar{y})}{\sqrt{\sum_{i=1}^{L} (x_i - \bar{x})^2 \cdot \sum_{i=1}^{L} (y_i - \bar{y})^2}}, \quad (14)$$

where $x_i, y_i \in \mathbb{R}$ run through all time points of the paired variates to be correlated. We then visualize the correlation between the variable features before and after HyperMamba. Fig. 5 shows that TimePro selects important time points through the time-tune strategy, which drives the learned multivariate correlations closer to the label sequences. This

suggests that TimePro effectively mitigates the detrimental effects of the multi-delay issue on accurate variable relationship modeling.

**Ablation of the feature dimensions** We first explore the effect of feature dimensions on forecasting performance, as shown in Fig. 6. For most datasets, once the feature dimension increases to a certain point (such as 48 or 64), further increasing the feature dimension may not significantly improve model performance, and can sometimes even lead to a decrease in performance. For example, the impact of feature dimensions on the Solar dataset is complex, with an initial increase in MAE (from 32 to 48), but a subsequent decrease in MAE when the dimensions are further increased (from 48 to 96). Therefore, selecting a moderate feature dimension is a better strategy. Moreover, TimePro achieves the best performance on most of the datasets when the feature dimension is 48. The small feature dimension ensures that TimePro is efficient.

**Ablation of the number of encoder layers** Regarding the number of encoder layers in Fig. 6, for the ECL dataset, increasing the number of encoder layers significantly improves performance, with the MAE gradually decreasing as the number of layers increases. For the Exchange dataset, increasing the number of layers significantly improves performance, but the improvement becomes smaller when the number of layers exceeds 2, eventually stabilizing at 3 and 4 layers. For the Weather dataset, increasing the number of layers has little impact on MAE, indicating that this dataset has a lower demand for the number of layers. For the Solar dataset, increasing the number of layers significantly improves MAE, especially when increasing from 1 to 3 layers, with a noticeable performance boost, but a slight increase at 4 layers. In summary, increasing the number of encoder layers generally improves model performance, but this improvement tends to level off or slow down after a certain point. For most datasets, an encoder layer count between 2 to 4 is likely to be a good choice, as it balances model complexity and performance.

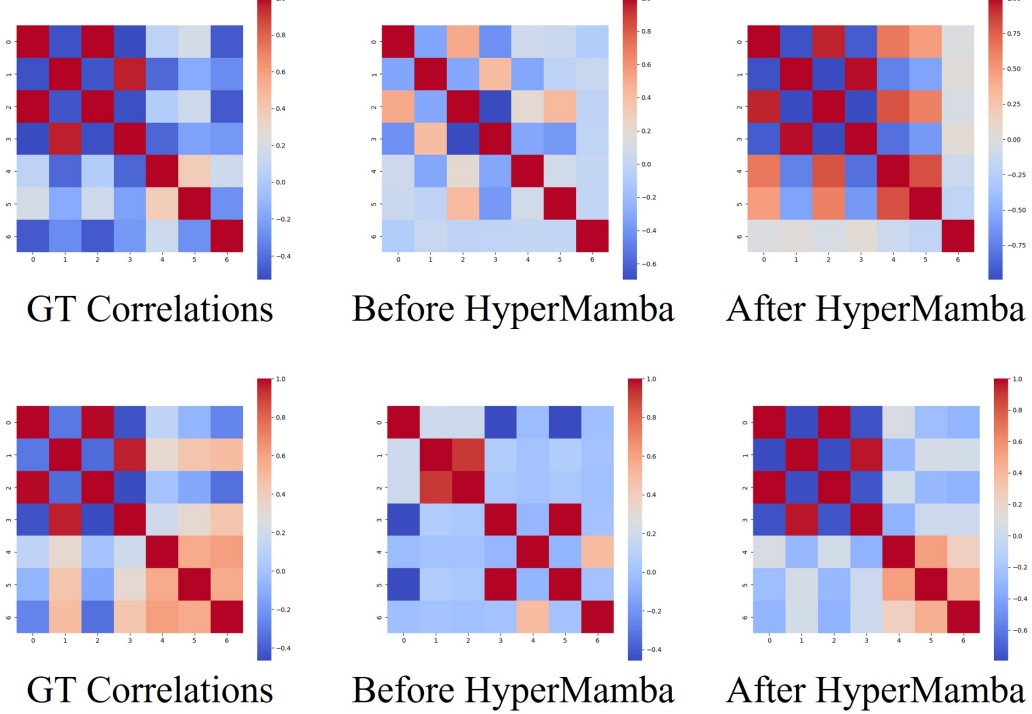

Figure 5: Visualization for multivariate correlation analysis on ETTm1 (upper) and ETTh1(bottom) dataset. The visualization is implemented based on the Pearson Correlation Coefficient. The GT Correlations denote the correlation between the variables of the forecast sequence (groundtruth). The two columns on the right denote the correlation between the variables before and after the HyperMamba module, respectively. It shows that TimePro drives the learned multivariate correlations closer to the forecast sequence through the HyperMamba module.

**Ablation of the patch length** Fig. 6 shows that the impact of patch length on model performance also varies from one dataset to another. For some datasets, increasing the patch length can significantly improve performance, while for others, the effect may be less pronounced. For the ECL dataset, increasing the patch length (from 8 to 32) reduces the MAE, with significant performance improvement due to patch length. For the Exchange dataset, the impact of patch length on MAE is relatively small, with only minor performance changes, indicating that this dataset has a lower demand for patch length. For the Weather dataset, the impact of patch length is more complex, with an initial increase in MAE (from 8 to 16), but a subsequent decrease in MAE when the patch length is further increased (from 16 to 32). For the Solar dataset, increasing the patch length from 8 to 32 significantly reduces the MAE, with clear performance improvement due to patch length. Overall, a moderate patch length (such as 16 to 32) may be a good starting point, as it can capture sufficient temporal sequence information without excessively increasing model complexity.

**Ablation of the time tune design** We next perform a further ablation analysis of the time tune strategy, as shown

Table 3: Ablation experiment of the time tune design on the Exchange and ETTh1 dataset. Results are averaged with horizon $H \in \{96, 192, 336, 720\}$.

| Variant | Exchange | | ETTh1 | |
|---|---|---|---|---|
| | MSE | MAE | MSE | MAE |
| Non-Adaptive | 0.360 | 0.402 | 0.451 | 0.441 |
| Adaptive | 0.352 | 0.399 | 0.438 | 0.438 |

in Table 3. An optional strategy is to integrate the variables states along the time dimension using a linear projection, i.e., non-adaptive. The results show that the non-adaptive variant obtains MSEs of 0.360 and 0.351 on Exchange and ETTh1, respectively. TimePro achieves smaller MSEs and MAEs on Exchange and ETTh1 compared to the non-adaptive variant. These experimental results validate that the introduction of an adaptive strategy can better tune the state of the variables and improve the prediction performance.

**Ablation of the HyperMamba structure** In addition, we also carry out an ablation analysis of HyperMamba's structure. Compared to the original Mamba, we remove two components, the depth-wise convolution before scanning

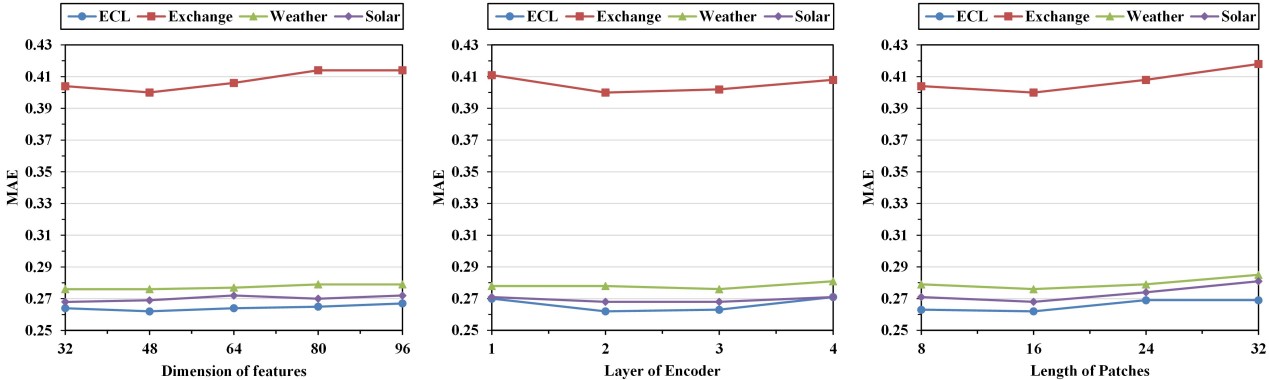

Figure 6: Influence of the hidden dimension of features D, Layer of encoder L and length of patches. We select ECL, Exchange, Weather and Solar-energy for visualization.

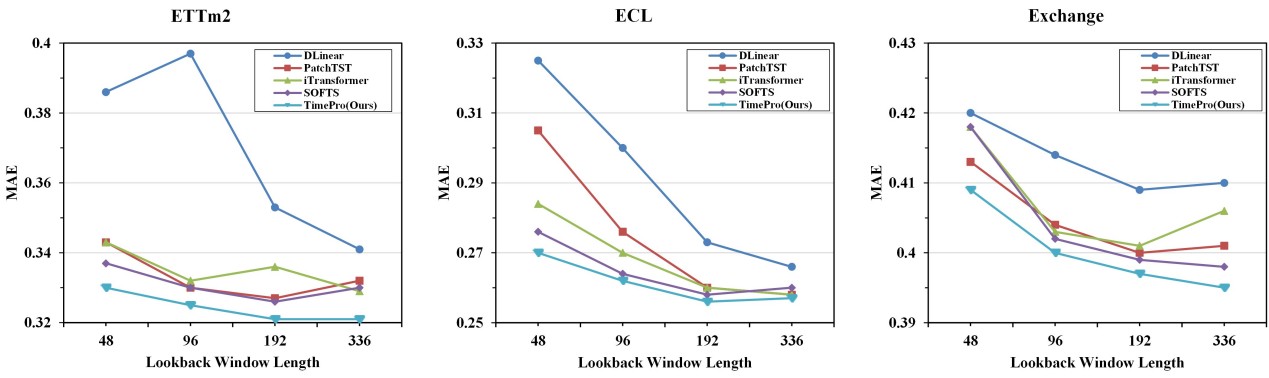

Figure 7: Influence of lookback window length $L \in 48, 96, 192, 336$ on ETTm2, ECL and Exchange dataset. TimePro performs almost consistently better than other models under different lookback window lengths.

Table 4: Ablation experiment of the HyperMamba structure on the Exchange and ETTh1 dataset. Results are averaged with horizon $H \in \{96, 192, 336, 720\}$.

| Variant | Exchange | | ETTh1 | |
|---|---|---|---|---|
| | MSE | MAE | MSE | MAE |
| Mamba + Hyper-Scan | 0.358 | 0.403 | 0.449 | 0.439 |
| - DWConv | 0.358 | 0.402 | 0.447 | 0.440 |
| - Linear | 0.356 | 0.401 | 0.447 | 0.441 |
| HyperMamba | 0.352 | 0.399 | 0.438 | 0.438 |

and the linear projection after scanning. As shown in Table 4, we replace the selective scanning module of the original Mamba with the Hyper-Scan and use it as the baseline, i.e., Mamba+Hyper-Scan. It can be observed that the forecasting performance is slightly improved after the removal of both components. We believe that the possible reason for this is that there is no localization between the variables and therefore depth-wise convolution is not necessary. In addition, the output linear within the original Mamba module is also redundant due to our introduction of TimeFFN. Therefore, we remove these two components, which could also improve the efficiency of TimePro.

## 6. Conclusion

In this paper, we begin by examining previous work on time series forecasting. These works tend to treat all variables or time points uniformly, which limits their ability to simultaneously capture complex variable relationships and obtain nontrivial temporal representations. Inspired by the recent popularity of state-space models (i.e., Mamba), we propose an efficient multivariate long-term time series forecasting model TimePro. In contrast to state transfer only in the variable dimension, TimePro adjusts the variable states through a time-tune strategy and uses the reconstructed hyper-states to obtain the output. The prediction performance is improved due to the ability of the hyperstates to perceive both complex variable relationships and intra-variable time variations. More remarkably, TimePro also absorbs the advantages of hardware awareness and linear complexity of the original mamba, thus ensuring high efficiency. Comprehensive experiments verify the effectiveness of TimePro. In future work, we plan to build a large time series prediction foundation model based on TimePro, thus providing the community with more directions to explore.

## Impact Statement

This paper presents work whose goal is to advance the field of Machine Learning. There are many potential societal consequences of our work, none which we feel must be specifically highlighted here.

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

## A. Dataset

We conduct experiments on eight real-world datasets. (1) ETT datasets: ETT (Zhou et al., 2021) (Electricity Transformer Temperature) comprises data on load and oil temperature, collected from electricity transformers over the period from July 2016 to July 2018. It contains four subsets, ETTm1, ETTm2, ETTh1 and ETTh2. ETTh1 and ETTh2 are recorded every hour, and ETTm1 and ETTm2 are recorded every 15 minutes. In addition, ETT datasets have few varieties and weak regularity. (2) Exchange (Wu et al., 2021) collects the panel data of daily exchange rates from 8 countries from 1990 to 2016. (3) Weather includes 21 indicators of weather, such as air temperature, and humidity. Its data is recorded every 10 min for 2020 in Germany. (4) Solar-Energy (Lai et al., 2018) records the solar power production of 137 PV plants in 2006, which is sampled every 10 minutes. (5) ECL (Wu et al., 2021) records the hourly electricity consumption data of 321 clients. The details of datasets are also provided in Table 5.

Table 5: Detailed dataset descriptions. *Channels* denotes the number of channels in each dataset. *Dataset Split* denotes the total number of time points in (Train, Validation, Test) split respectively. *Prediction Length* denotes the future time points to be predicted and four prediction settings are included in each dataset. *Granularity* denotes the sampling interval of time points.

| Dataset | Channels | Prediction Length | Dataset Split | Granularity | Domain |
|---|---|---|---|---|---|
| ETTh1, ETTh2 | 7 | {96, 192, 336, 720} | (8545, 2881, 2881) | Hourly | Electricity |
| ETTm1, ETTm2 | 7 | {96, 192, 336, 720} | (34465, 11521, 11521) | 15min | Electricity |
| Weather | 21 | {96, 192, 336, 720} | (36792, 5271, 10540) | 10min | Weather |
| ECL | 321 | {96, 192, 336, 720} | (18317, 2633, 5261) | Hourly | Electricity |
| Exchange | 8 | {96, 192, 336, 720} | (5120, 665, 1422) | Daily | Economy |
| Solar-Energy | 137 | {96, 192, 336, 720} | (36601, 5161, 10417) | 10min | Energy |

## B. Implementation Details

### B.1. Details of time and variable preserved embedding

Following PatchTST (Nie et al., 2023), we divide each input univariate time series $X_{i,:} \in \mathbb{R}^L$ into patches that can overlap. Let the patch length be $P_l$ and the stride between two consecutive patches be $S_l$. The patching process generates a sequence of patches $X_{i,:}^p \in \mathbb{R}^{P \times D}$, where $P$ is the number of patches and $P = \lfloor \frac{L - P_l}{S_l} \rfloor + 2$. Here, we fill the last value of $X_{i,:} \in \mathbb{R}^L$ with the number of $S_l$ repetitions to the end of the original sequence and then patch it. By using patching, the number of input tokens can be reduced from $L$ to approximately $\frac{L}{S_l}$. This means that TimePro's memory usage and computational complexity are reduced by a factor of $S_l$.

### B.2. Experiment Details

All experiments are performed on 4 NVIDIA Tesla V100 GPUs with 32G VRAM. The model is optimized using the Mean Squared Error (MSE) loss function. The performance of the different methods is compared based on two main evaluation metrics: the Mean Squared Error (MSE) and the Mean Absolute Error (MAE). We use the ADAM optimizer (Kingma & Ba, 2017) with an initial learning rate of $5 \times 10$-4. This rate is modulated by a cosine learning rate scheduler. We study the layer of Problocks $\gamma$ within the set {1,2,3,4} and the dimension of the embedding $D$ within {32,48,64,96}.

## C. Full Results

Table 6 shows the full results of the prediction benchmarks. We conduct experiments using eight widely used real-world datasets and compare our method to ten previous state-of-the-art models. The comparison models use a variety of architectures including convolution, transformer, mlp, and mamba. In these tests, our TimePro exhibits strong performance. Specifically, TimePro achieves the best performance on most of the datasets. Both at different prediction lengths and on average, TimePro possesses smaller MSE and MAE.

In addition, we implement visualizations on the ETTh2 and ECL dataset to further validate the predictive effect of TimePro.

As shown in Fig. 8 and 9, compared to iTransformer and S-Mamba, TimePro's prediction curves are closer to the groundtruth, and the curve distances are smaller at the inflection point locations. The visualization analysis further validates the effectiveness of TimePro.

Table 6: Multivariate long-term forecasting results with prediction lengths $H \in \{96, 192, 336, 720\}$ and fixed lookback window length $L = 96$. The results are are taken from iTransformer (Liu et al., 2024b) and S-Mamba (Wang et al., 2025).

| Models | | TimePro ours | | S-Mamba (2025) | | SOFTS (2024) | | iTransformer (2024b) | | PatchTST (2023) | | Crossformer (2023) | | TiDE (2023) | | TimesNet (2023) | | DLinear (2023) | | SCINet (2022) | | Stationary (2022) | |
|---|---|---|---|---|---|---|---|---|---|---|---|---|---|---|---|---|---|---|---|---|---|---|---|
| Metric | | MSE | MAE | MSE | MAE | MSE | MAE | MSE | MAE | MSE | MAE | MSE | MAE | MSE | MAE | MSE | MAE | MSE | MAE | MSE | MAE | MSE | MAE |
| ETTm1 | 96 | 0.326 | 0.364 | 0.333 | 0.368 | 0.325 | 0.361 | 0.334 | 0.368 | 0.329 | 0.365 | 0.404 | 0.426 | 0.364 | 0.387 | 0.338 | 0.375 | 0.345 | 0.372 | 0.418 | 0.438 | 0.386 | 0.398 |
| | 192 | 0.367 | 0.383 | 0.376 | 0.390 | 0.375 | 0.389 | 0.377 | 0.391 | 0.380 | 0.394 | 0.450 | 0.451 | 0.398 | 0.404 | 0.374 | 0.387 | 0.380 | 0.389 | 0.439 | 0.450 | 0.459 | 0.444 |
| | 336 | 0.402 | 0.409 | 0.408 | 0.413 | 0.405 | 0.412 | 0.426 | 0.420 | 0.400 | 0.410 | 0.532 | 0.515 | 0.428 | 0.425 | 0.410 | 0.411 | 0.413 | 0.413 | 0.490 | 0.485 | 0.495 | 0.464 |
| | 720 | 0.469 | 0.446 | 0.475 | 0.448 | 0.466 | 0.447 | 0.491 | 0.459 | 0.475 | 0.453 | 0.666 | 0.589 | 0.487 | 0.461 | 0.478 | 0.450 | 0.474 | 0.453 | 0.595 | 0.550 | 0.585 | 0.516 |
| | Avg | 0.391 | 0.400 | 0.398 | 0.405 | 0.393 | 0.403 | 0.407 | 0.410 | 0.396 | 0.406 | 0.513 | 0.496 | 0.419 | 0.419 | 0.400 | 0.406 | 0.403 | 0.407 | 0.485 | 0.481 | 0.481 | 0.456 |
| ETTm2 | 96 | 0.178 | 0.260 | 0.179 | 0.263 | 0.180 | 0.261 | 0.180 | 0.264 | 0.184 | 0.264 | 0.287 | 0.366 | 0.207 | 0.305 | 0.187 | 0.267 | 0.193 | 0.292 | 0.286 | 0.377 | 0.192 | 0.274 |
| | 192 | 0.242 | 0.303 | 0.250 | 0.309 | 0.246 | 0.306 | 0.250 | 0.309 | 0.246 | 0.306 | 0.414 | 0.492 | 0.290 | 0.364 | 0.249 | 0.309 | 0.284 | 0.362 | 0.399 | 0.445 | 0.280 | 0.339 |
| | 336 | 0.303 | 0.342 | 0.312 | 0.349 | 0.319 | 0.352 | 0.311 | 0.348 | 0.308 | 0.346 | 0.597 | 0.542 | 0.377 | 0.422 | 0.321 | 0.351 | 0.369 | 0.427 | 0.637 | 0.591 | 0.334 | 0.361 |
| | 720 | 0.400 | 0.399 | 0.411 | 0.406 | 0.405 | 0.401 | 0.412 | 0.407 | 0.409 | 0.402 | 1.730 | 1.042 | 0.558 | 0.524 | 0.408 | 0.403 | 0.554 | 0.522 | 0.960 | 0.735 | 0.417 | 0.413 |
| | Avg | 0.281 | 0.326 | 0.288 | 0.332 | 0.287 | 0.330 | 0.288 | 0.332 | 0.287 | 0.330 | 0.757 | 0.610 | 0.358 | 0.404 | 0.291 | 0.333 | 0.350 | 0.401 | 0.571 | 0.537 | 0.306 | 0.347 |
| ETTh1 | 96 | 0.375 | 0.398 | 0.386 | 0.405 | 0.381 | 0.399 | 0.386 | 0.405 | 0.394 | 0.406 | 0.423 | 0.448 | 0.479 | 0.464 | 0.384 | 0.402 | 0.386 | 0.400 | 0.654 | 0.599 | 0.513 | 0.491 |
| | 192 | 0.427 | 0.429 | 0.443 | 0.437 | 0.435 | 0.431 | 0.441 | 0.436 | 0.440 | 0.435 | 0.471 | 0.474 | 0.525 | 0.492 | 0.436 | 0.429 | 0.437 | 0.432 | 0.719 | 0.631 | 0.534 | 0.504 |
| | 336 | 0.472 | 0.450 | 0.489 | 0.468 | 0.480 | 0.452 | 0.487 | 0.458 | 0.491 | 0.462 | 0.570 | 0.546 | 0.565 | 0.515 | 0.491 | 0.469 | 0.481 | 0.459 | 0.778 | 0.659 | 0.588 | 0.535 |
| | 720 | 0.476 | 0.474 | 0.502 | 0.489 | 0.499 | 0.488 | 0.503 | 0.491 | 0.487 | 0.479 | 0.653 | 0.621 | 0.594 | 0.558 | 0.521 | 0.500 | 0.519 | 0.516 | 0.836 | 0.699 | 0.643 | 0.616 |
| | Avg | 0.438 | 0.438 | 0.455 | 0.450 | 0.449 | 0.442 | 0.454 | 0.447 | 0.453 | 0.446 | 0.529 | 0.522 | 0.541 | 0.507 | 0.458 | 0.450 | 0.456 | 0.452 | 0.747 | 0.647 | 0.570 | 0.537 |
| ETTh2 | 96 | 0.293 | 0.345 | 0.296 | 0.348 | 0.297 | 0.347 | 0.297 | 0.349 | 0.288 | 0.340 | 0.745 | 0.584 | 0.400 | 0.440 | 0.340 | 0.374 | 0.333 | 0.387 | 0.707 | 0.621 | 0.476 | 0.458 |
| | 192 | 0.367 | 0.394 | 0.376 | 0.396 | 0.373 | 0.394 | 0.380 | 0.400 | 0.376 | 0.395 | 0.877 | 0.656 | 0.528 | 0.509 | 0.402 | 0.414 | 0.477 | 0.476 | 0.860 | 0.689 | 0.512 | 0.493 |
| | 336 | 0.419 | 0.431 | 0.424 | 0.431 | 0.410 | 0.426 | 0.428 | 0.432 | 0.440 | 0.451 | 1.043 | 0.731 | 0.643 | 0.571 | 0.452 | 0.452 | 0.594 | 0.541 | 1.000 | 0.744 | 0.552 | 0.551 |
| | 720 | 0.427 | 0.445 | 0.426 | 0.444 | 0.411 | 0.433 | 0.427 | 0.445 | 0.436 | 0.453 | 1.104 | 0.763 | 0.874 | 0.679 | 0.462 | 0.468 | 0.831 | 0.657 | 1.249 | 0.838 | 0.562 | 0.560 |
| | Avg | 0.377 | 0.403 | 0.381 | 0.405 | 0.373 | 0.400 | 0.383 | 0.407 | 0.385 | 0.410 | 0.942 | 0.684 | 0.611 | 0.550 | 0.414 | 0.427 | 0.559 | 0.515 | 0.954 | 0.723 | 0.526 | 0.516 |
| ECL | 96 | 0.139 | 0.234 | 0.139 | 0.235 | 0.143 | 0.233 | 0.148 | 0.240 | 0.164 | 0.251 | 0.219 | 0.314 | 0.237 | 0.329 | 0.168 | 0.272 | 0.197 | 0.282 | 0.247 | 0.345 | 0.169 | 0.273 |
| | 192 | 0.156 | 0.249 | 0.159 | 0.255 | 0.158 | 0.248 | 0.162 | 0.253 | 0.173 | 0.262 | 0.231 | 0.322 | 0.236 | 0.330 | 0.184 | 0.289 | 0.196 | 0.285 | 0.257 | 0.355 | 0.182 | 0.286 |
| | 336 | 0.172 | 0.267 | 0.176 | 0.272 | 0.178 | 0.269 | 0.178 | 0.269 | 0.190 | 0.279 | 0.246 | 0.337 | 0.249 | 0.344 | 0.198 | 0.300 | 0.209 | 0.301 | 0.269 | 0.369 | 0.200 | 0.304 |
| | 720 | 0.209 | 0.299 | 0.204 | 0.298 | 0.218 | 0.305 | 0.225 | 0.317 | 0.230 | 0.313 | 0.280 | 0.363 | 0.284 | 0.373 | 0.220 | 0.320 | 0.245 | 0.333 | 0.299 | 0.390 | 0.222 | 0.321 |
| | Avg | 0.169 | 0.262 | 0.170 | 0.265 | 0.174 | 0.264 | 0.178 | 0.270 | 0.189 | 0.276 | 0.244 | 0.334 | 0.251 | 0.344 | 0.192 | 0.295 | 0.212 | 0.300 | 0.268 | 0.365 | 0.193 | 0.296 |
| Exchange | 96 | 0.085 | 0.204 | 0.086 | 0.207 | 0.091 | 0.209 | 0.086 | 0.206 | 0.088 | 0.205 | 0.256 | 0.367 | 0.094 | 0.218 | 0.107 | 0.234 | 0.088 | 0.218 | 0.267 | 0.396 | 0.111 | 0.237 |
| | 192 | 0.178 | 0.299 | 0.182 | 0.304 | 0.176 | 0.303 | 0.177 | 0.299 | 0.176 | 0.299 | 0.470 | 0.509 | 0.184 | 0.307 | 0.226 | 0.344 | 0.176 | 0.315 | 0.351 | 0.459 | 0.219 | 0.335 |
| | 336 | 0.328 | 0.414 | 0.332 | 0.418 | 0.329 | 0.416 | 0.331 | 0.417 | 0.301 | 0.397 | 1.268 | 0.883 | 0.349 | 0.431 | 0.367 | 0.448 | 0.313 | 0.427 | 1.324 | 0.853 | 0.421 | 0.476 |
| | 720 | 0.817 | 0.679 | 0.867 | 0.703 | 0.848 | 0.680 | 0.847 | 0.691 | 0.901 | 0.714 | 1.767 | 1.068 | 0.852 | 0.698 | 0.964 | 0.746 | 0.839 | 0.695 | 1.058 | 0.797 | 1.092 | 0.769 |
| | Avg | 0.352 | 0.399 | 0.367 | 0.408 | 0.361 | 0.402 | 0.360 | 0.403 | 0.367 | 0.404 | 0.940 | 0.707 | 0.370 | 0.413 | 0.416 | 0.443 | 0.354 | 0.414 | 0.750 | 0.626 | 0.461 | 0.454 |
| Weather | 96 | 0.166 | 0.207 | 0.165 | 0.210 | 0.166 | 0.208 | 0.174 | 0.214 | 0.176 | 0.217 | 0.158 | 0.230 | 0.202 | 0.261 | 0.172 | 0.220 | 0.196 | 0.255 | 0.221 | 0.306 | 0.173 | 0.223 |
| | 192 | 0.216 | 0.254 | 0.214 | 0.252 | 0.217 | 0.253 | 0.221 | 0.254 | 0.221 | 0.256 | 0.206 | 0.277 | 0.242 | 0.298 | 0.219 | 0.261 | 0.237 | 0.296 | 0.261 | 0.340 | 0.245 | 0.285 |
| | 336 | 0.273 | 0.296 | 0.274 | 0.296 | 0.282 | 0.300 | 0.278 | 0.296 | 0.275 | 0.296 | 0.272 | 0.335 | 0.287 | 0.335 | 0.280 | 0.306 | 0.283 | 0.335 | 0.309 | 0.378 | 0.321 | 0.338 |
| | 720 | 0.351 | 0.346 | 0.350 | 0.345 | 0.356 | 0.351 | 0.358 | 0.347 | 0.352 | 0.346 | 0.398 | 0.418 | 0.351 | 0.386 | 0.365 | 0.359 | 0.345 | 0.381 | 0.377 | 0.427 | 0.414 | 0.410 |
| | Avg | 0.251 | 0.276 | 0.251 | 0.276 | 0.255 | 0.278 | 0.258 | 0.278 | 0.256 | 0.279 | 0.259 | 0.315 | 0.271 | 0.320 | 0.259 | 0.287 | 0.265 | 0.317 | 0.292 | 0.363 | 0.288 | 0.314 |
| Solar-Energy | 96 | 0.196 | 0.237 | 0.205 | 0.244 | 0.200 | 0.230 | 0.203 | 0.237 | 0.205 | 0.246 | 0.310 | 0.331 | 0.312 | 0.399 | 0.250 | 0.292 | 0.290 | 0.378 | 0.237 | 0.344 | 0.215 | 0.249 |
| | 192 | 0.231 | 0.263 | 0.237 | 0.270 | 0.229 | 0.253 | 0.233 | 0.261 | 0.237 | 0.267 | 0.734 | 0.725 | 0.339 | 0.416 | 0.296 | 0.318 | 0.320 | 0.398 | 0.280 | 0.380 | 0.254 | 0.272 |
| | 336 | 0.250 | 0.281 | 0.258 | 0.288 | 0.243 | 0.269 | 0.248 | 0.273 | 0.250 | 0.276 | 0.750 | 0.735 | 0.368 | 0.430 | 0.319 | 0.330 | 0.353 | 0.415 | 0.304 | 0.389 | 0.290 | 0.296 |
| | 720 | 0.253 | 0.285 | 0.260 | 0.288 | 0.245 | 0.272 | 0.249 | 0.275 | 0.252 | 0.275 | 0.769 | 0.765 | 0.370 | 0.425 | 0.338 | 0.337 | 0.356 | 0.413 | 0.308 | 0.388 | 0.285 | 0.200 |
| | Avg | 0.232 | 0.266 | 0.240 | 0.273 | 0.229 | 0.256 | 0.233 | 0.262 | 0.236 | 0.266 | 0.641 | 0.639 | 0.347 | 0.417 | 0.301 | 0.319 | 0.330 | 0.401 | 0.282 | 0.375 | 0.261 | 0.381 |

# D. More ablation results

**Ablation of the Hyper-Scan**  We explore the effectiveness of hyper-scan as shown in Table 7. Specifically, we remove the HyperMamba module as the baseline and set up several variants, including scanning in the variable dimension only (Var only), scanning in the time dimension only (Time only), and scanning in the variable dimension first and then in the time dimension (Time & Var). The results show that both time only and var only improve the prediction performance of the baseline, which is also attributed to the powerful global dependency capture of the original Mamba. In addition, the combination of the two further reduces the MSE and MAE of the prediction. However, this structure does not model the relationship between time and variables in a single module, and thus performance is limited. In addition, it can lead to large computational costs that impair the efficiency of the model. In contrast, TimePro models more accurate variable dependencies through an efficient time tune strategy, requires only linear computational complexity, and achieves the best prediction performance.

We also provide some visualization results to validate the effectiveness of Hyper-Scan, as shown in Fig. 10. It can be observed that suboptimal results are obtained for the prediction curves of models that scan only in the time dimension or the variable dimension. Specifically, when scanning only in the variable dimension (Fig. a), the model's prediction curves are smoother, with a poorer fit at the extremes. We attribute this to the model's lack of ability to capture the details of local changes within variables. When scanning in the time dimension only (Fig. b), the model's prediction ability for the extremes improves, but the average accuracy remains poor. This is due to the lack of capturing complex variable relationships. And when we use non-adaptive hyper-scanning (Fig. c), the model's average accuracy and predictive ability for extreme values are improved, but the performance is still unsatisfactory. And when we apply the adaptive hyper-scan, i.e., TimePro (Fig. d), the model can perceive both variable relationships and significant temporal information, resulting in a more accurate prediction performance.

Table 7: Ablation experiment of the Hyper-Scan on the Exchange and ETTh1 dataset. Results are averaged with horizon $H \in \{96, 192, 336, 720\}$.

| Variant | Exchange | | ETTh1 | |
|---|---|---|---|---|
| | MSE | MAE | MSE | MAE |
| Baseline | 0.374 | 0.418 | 0.467 | 0.452 |
| Time only | 0.367 | 0.414 | 0.459 | 0.447 |
| Var only | 0.367 | 0.410 | 0.454 | 0.443 |
| Time & Var | 0.360 | 0.404 | 0.450 | 0.439 |
| TimePro | 0.352 | 0.399 | 0.438 | 0.438 |

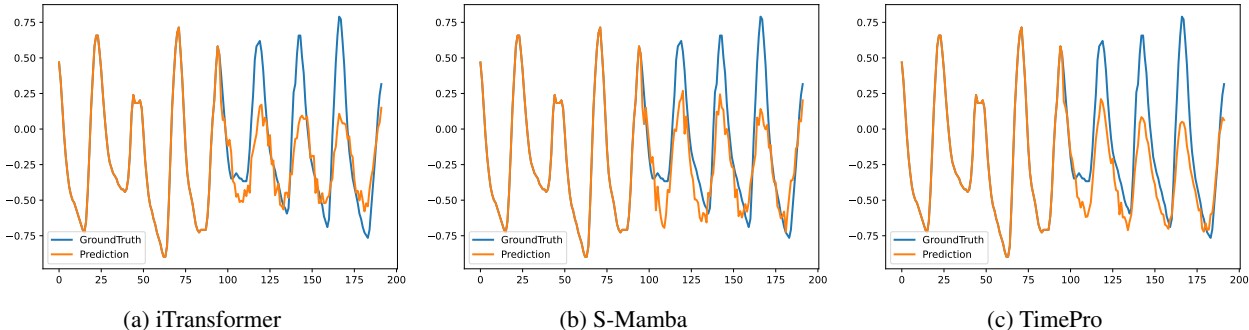

(a) iTransformer          (b) S-Mamba          (c) TimePro

Figure 8: Comparison of forecasts between TimePro, S-Mamba and iTransformer on ETTh2 dataset when the input length is 96 and the forecast length is 96. The blue line represents the ground truth and the orange line represents the forecast.

**Efficiency effect of variable channels** We further explore the efficiency of TimePro with different variable channels in Fig. 11, which can be seen as a complement to Table 1. It can be observed that as the number of variable channels increases, iTransformer shows a quadratic increase in both the memory and inference time, which severely compromises the efficiency of the model and limits practical applications. In addition, PatchTST has unsatisfactory efficiency under all variable channels. In contrast, the linear scaling ability compared to variable channels, small memory consumption and high inference speed validate TimePro's efficiency.

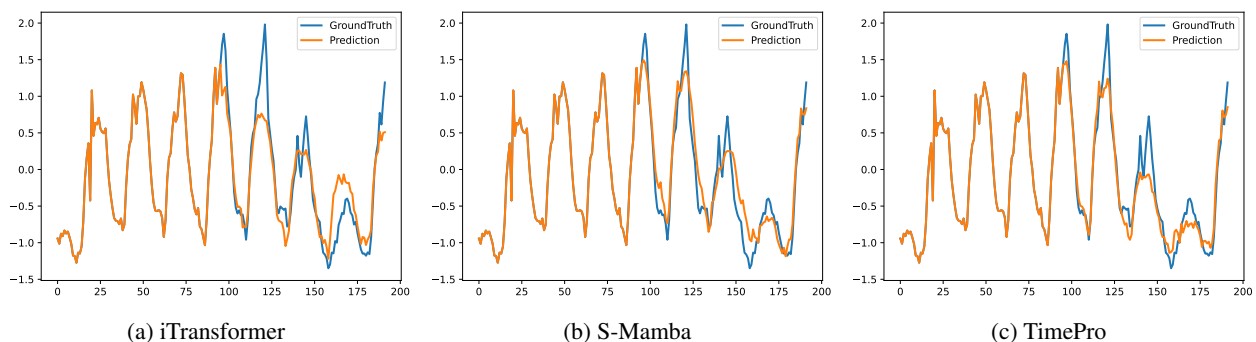

(a) iTransformer       (b) S-Mamba       (c) TimePro

Figure 9: Comparison of forecasts between TimePro, S-Mamba and iTransformer on ECL dataset when the input length is 96 and the forecast length is 96. The blue line represents the ground truth and the orange line represents the forecast.

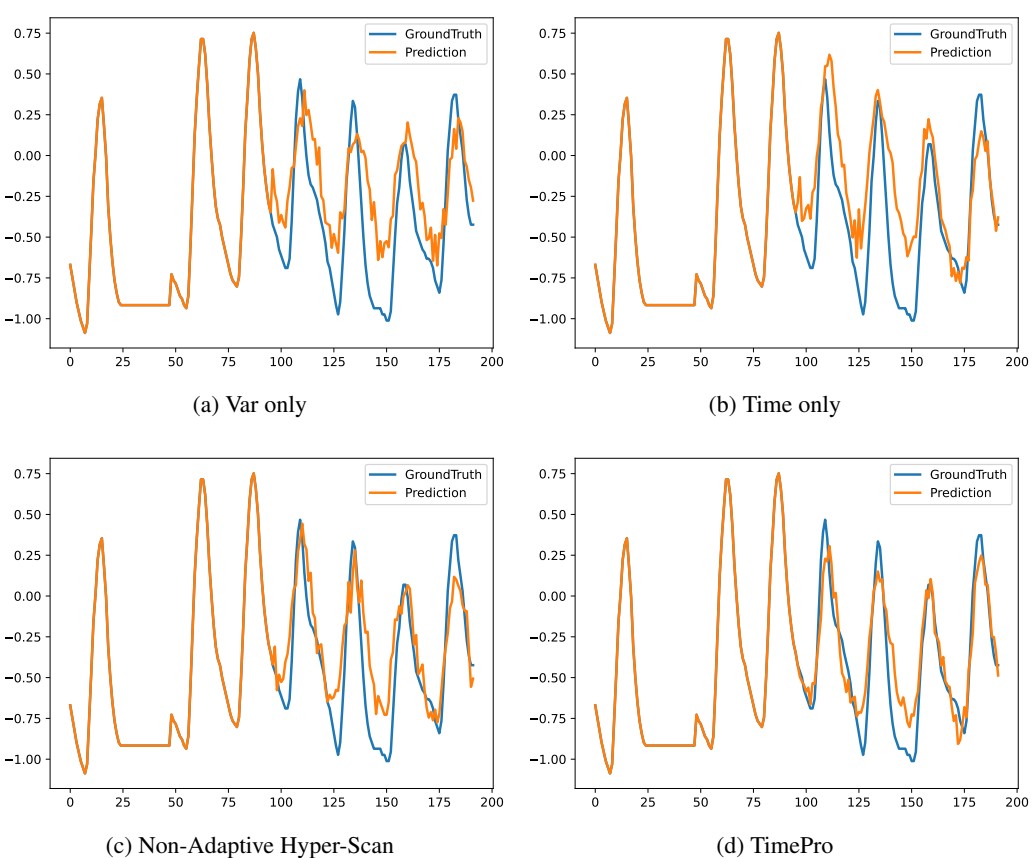

(a) Var only       (b) Time only

(c) Non-Adaptive Hyper-Scan       (d) TimePro

Figure 10: Visual comparison of different Hyper-scan designs on ETTh2 dataset when the input length is 96 and the forecast length is 96. Var only and Time only denote we only apply the scanning in the variable and time dimension, respectively. Non-Adaptive Hyper-Scan denote we integrate the variables states along the time dimension using a linear projection, i.e., non-adaptive. The better prediction curves of TimePro prove the validity of Hyper-Scan's structure.

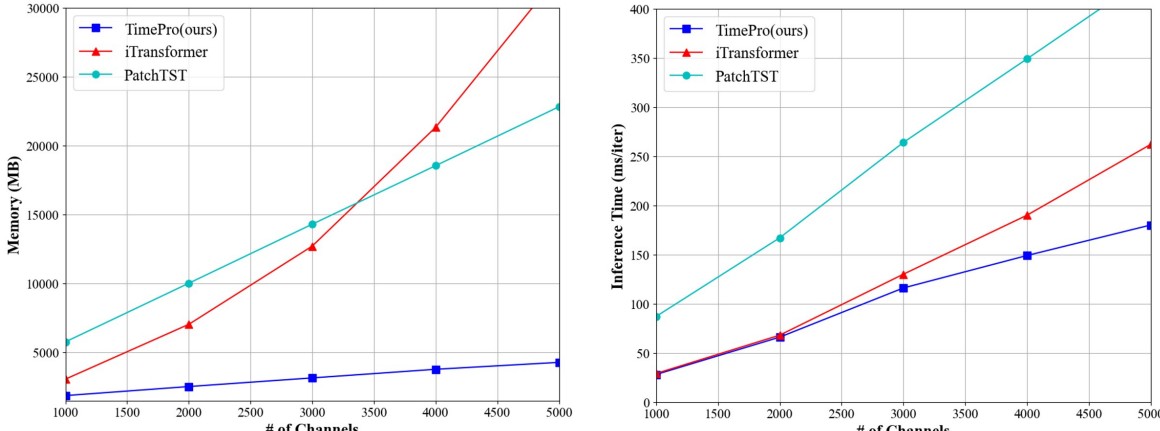

Figure 11: Memory and inference time of different methods. We set the lookback window L = 96, forecast horizon H = 720 in a synthetic dataset we conduct. We set the batch size to 4 due to the limited GPU memory. The inference times are measured on Nvidia V100 GPU. It reveals that TimePro scales to large number of channels more effectively than Transformer-based models (i.e., iTransformer and PatchTST).

