# OpenReview forum: "TimePro: Efficient Multivariate Long-term Time Series Forecasting with Variable- and Time-Aware Hyper-state"
_ICML.cc/2025/Conference — ICML 2025 poster_

### Official Review · Reviewer_1d3n · 2025-03-09

**Overall Recommendation:** 3

**Summary:**

This paper introduces TimePro, a model designed for multivariate long-term time series forecasting, but it is marred by significant writing issues. The main problem lies in the overly complex and unclear explanations. The terminology used, such as "variable- and time-aware hyper-states," is vague and confusing, making the methodology difficult to follow, especially for readers not already familiar with the specific framework. The paper overcomplicates simple ideas with jargon that could have been explained more clearly, detracting from its overall accessibility and readability. Additionally, there is a lack of coherence between sections, with abrupt shifts in focus that make the paper feel disjointed. The presentation of the model and its components lacks clarity, and the descriptions of the model's workings are not intuitive, making it harder for readers to understand the core innovations. The overall structure feels cumbersome, and a more straightforward and concise approach would significantly improve the readability and impact of the paper.

## Update after rebuttal
I checked the rebuttal, and most of my concerns are addressed. Therefore, I raise my score from 2 to 3.

**Claims And Evidence:**

The claim of this paper are not supported by clear evidence. See "Other Strengths and Weaknesses" for details.

**Essential References Not Discussed:**

N/A

**Experimental Designs Or Analyses:**

The paper includes some ablation experiments; however, due to the lack of motivation behind the module design, these ablation experiments appear meaningless.

**Methods And Evaluation Criteria:**

The proposed method simply applies Mamba to time series forecasting, and the improvements in various modules lack motivation.

**Other Comments Or Suggestions:**

See "Other Strengths and Weaknesses".

**Other Strengths And Weaknesses:**

## Strengths：

The paper presents extensive experiments on several real-world datasets, validating the model’s robustness across different domains and prediction horizons.

## Weakness:

1. The "multi-delay issue" should be given a strict definition rather than a simple description, as it is the core problem addressed by the paper.

2. The authors claim in the abstract, "Traditional models typically process all variables or time points uniformly, which limits their ability to capture complex variable relationships and obtain non-trivial time representations." How does this statement logically connect to the previously mentioned "multi-delay issue"? It feels like there is a lack of logical linkage between these two sentences, making the paragraph appear confusing.

3. The description of the method in the introduction and abstract does not seem to address the "multi-delay issue" proposed by the authors, which makes the proposed method seem meaningless.

4. In the first and second paragraphs of the introduction, a large portion of the content focuses on the Mamba model, which is not directly related to the research problem (time series forecasting). This deviates from the main topic and wastes space that should focus on the challenges of time series forecasting and the contributions of this research. While Mamba provides relevant background value, the excessive detail introduced does not directly relate to the core issue of this paper and may confuse readers, affecting the overall coherence and academic rigor. The authors should consider streamlining this content and focusing more on the research background that is directly related to the paper's goals, thus better highlighting its innovation and practical significance.

5. Why introduce Transformer-based time series forecasting methods in the third paragraph of the introduction when this paper is based on a Mamba-based time series forecasting method?

6 . When introducing "Preliminaries" in Section 3, please first clarify what the inputs and outputs are, as this will prevent confusion for readers who are not familiar with Mamba and time series forecasting.

7. The method section of the paper reads like code documentation, lacking explanations of the motivations behind the module design.

8 . The experimental section spends nearly 80% of its space on ablation experiments, leaving very little room for other important information. This makes the experimental section seem sparse.

**Questions For Authors:**

See "Other Strengths and Weaknesses".

**Relation To Broader Scientific Literature:**

This paper merely applies the Mamba tool to time series forecasting, making the paper seem incremental.

**Theoretical Claims:**

The paper does not contain any theoretical claims.

---

> ### Author Rebuttal · Authors · 2025-04-01
>
> We sincerely thank you for your time and efforts in reviewing our work and providing thoughtful feedback that can further strengthen our manuscript. We have added some experiments following your suggestions and they are available at  https://anonymous.4open.science/r/Anonymous_figure-2319/figure_response.pdf. Please see our detailed responses to your comments below.
> ### 1 Multi-delay issue definition
> We have defined the multi-delay issue in both the abstract and paragraph 3 of the introduction. To make this definition strict, we modify the definition as follows.
> The multi-delay issue in multivariate time series forecasting is defined as the temporal discrepancy in the propagation of influence from different predictor variables to the target variable, characterized by the presence of distinct and non-uniform time lags between the changes in predictor variables and their corresponding effects on the target variable.
> ### 2 Logical connection
> To effectively address the multi-delay issue, it is necessary for the model to identify and capture the critical temporal points of each variable within the input time series data. However, traditional models uniformly process different time points of the same variable, thereby causing the key temporal information to be obscured by a vast number of irrelevant time series features. Therefore, the traditional model can not solve the multi-delay issue. We will revise the above content into the paper.
> ### 3 Addressing multi-delay issue
> We have detailed how TimePro solves the multi-delay problem in paragraph 4 of introduction. During the scan of mamba, a specialized network is employed to learn the offsets of critical time points. By adaptively selecting these key time points, TimePro dynamically updates the hidden states to reflect the most salient temporal information. Through adaptive selection of key temporal features, the model focuses more on the delay times of each variable, thus solving the multi-delay problem.
> ### 4 Excessive Mamba detail
> I'm sorry for your concerns. Taking your suggestion into consideration, we will revise the first section to mainly focus on the advantages of Mamba in time-series forecasting, thereby better aligning with the theme of time-series forecasting. In the second paragraph we introduce some Mamba-based works that is relevant to our approach, which can facilitate the reader to understand our contribution and differences from previous work. We will simplify the descriptions of these methods and enhance the summarization of the methods in relation to the theme. We will revise the last sentence of the second paragraph as follows: The mentioned related works often employ different ways to scan features from various directions. However, these methods overlook the fact that different variables have different impact durations on the target variable, which limits their performance.
> ### 5 Reason of Transformer-based methods introduction
> Although TimePro is based on Mamba, transformer-based models dominate the time series community. Therefore, we need to describe these models in the introduction and motivate the reader to understand the shortcomings of these models. As we mention in lines 59-66 of our paper, transformer-based models also can’t capture critical time points. We believe that using some space to describe this work is necessary to help make our motivation clearer.
> ### 6 Clarify inputs and outputs
> The inputs and outputs of the model refer to arbitrary sequences, as shown in lines 145-146 of our paper. Considering the complex mechanism of Mamba, we need to depict the working mechanism of Mamba before introducing our TimePro (i.e., Preliminaries), which facilitates the understanding of our approach. This writing structure is also adopted by other Mamba-based models, including VMamba (NeurIPS2024), ViM (ICML2024) and S-Mamba.
> ### 7 Lacking explanations of the motivations
> We present our motivation before introducing our core modules (HyperMamba and Hyper-Scan), as shown in lines 227-241 and lines 220-229. In addition, we also introduce the motivation for Time and variable preserved embedding, as shown in lines 191-197 of our paper. For other parts such as ProBlock and Linear projection, we do not present the motivation behind them as they do not involve architectural innovations. We will be more specific in describing the motivation before each module description in the final version.
> ### 8 Supplementary experiments
> We have provided some other experimental results in the figure response, including R1: Efficiency comparison, R2: Memory and inference time with different channels, etc. In addition, we will put Fig. R1 and R2 into the final version to increase the space of the comparison experiment. Other experiments will be placed in the appendix.
>
> We will follow your suggestions for the writing to reorganize our final version. If the rebuttal helps address your concerns, we kindly ask that you increase your score to give TimePro a fighting chance!

---

> > ### Comment · Reviewer_1d3n · 2025-04-03
> >
> > Thank you for the detailed rebuttal. Unfortunately, I still have the following concerns.
> >
> > **Definition of the multi-delay issue**
> >
> > The concern is not merely about how the issue is defined in words. If this is a widely acknowledged problem in the field, then proper citations to prior work are expected. On the other hand, if this issue is first proposed by your paper, then there should be clear evidence demonstrating its existence — either through compelling experimental results or theoretical justification.
> >
> > **Logical connection**
> >
> > Thank you for the clarification. This is not a major concern.
> >
> > **Addressing the issue**
> >
> > Again, thank you for the clarification. However, similar to the concern above, is there any justification (e.g., controlled experiments, ablation studies, or theoretical analysis) that demonstrates your method’s effectiveness in specifically addressing the multi-delay issue?
> >
> > **Excessive detail on Mamba and input/output description**
> >
> > These are structural issues. I believe the paper will improve after revision.
> >
> > **Explanation of motivations**
> >
> > This connects back to the first concern. The key point is to first establish that the issue exists, and then clearly explain how your method addresses it.
> >
> > **Supplementary experiments**
> >
> > This is not a major concern.
> >
> >
> > Overall, my main concern lies in the lack of justification for the existence of the multi-delay issue, and whether TimePro effectively addresses this issue in a demonstrable way.
> >
> > # Reply to Reply Rebuttal Comment by Authors
> >
> > Thanks for addressing my concerns. I'm raising my score to 3 (Weak Accept). Figure R7 is intuitive and should be provided in the revised version. Good luck and hope you well.

---

> > > ### Author Response · Authors · 2025-04-07
> > >
> > > Thank you very much for your time devoted to reviewing our paper and your constructive comments. After previous discussions, we notice that you are confused about the existence of the multi-delay issue and how we can justify solving it. We have added some experiments following your suggestions and they are available at https://anonymous.4open.science/r/Anonymous_figure-2319/figure_response.pdf.  Then, Please see our detailed responses to your comments below.
> > >
> > > ## Existence of the multi-delay issue
> > > The multi-delay issue is a fundamental research topic in the multivariate time series forecasting. The multi-delay problem is mentioned in references [1-3]. We apologize for the confusion caused by not citing these papers. We will follow your comments and add some references [1-3] to the revised version.
> > >
> > > [1] Detecting time lag between a pair of time series using visibility graph algorithm.
> > >
> > > [2] Lag penalized weighted correlation for time series clustering
> > >
> > > [3] Multivariate time delay analysis based local KPCA fault prognosis approach for nonlinear processes
> > > ## TimePro effectively addresses multi-delay issue in a demonstrable way
> > > We demonstrate that TimePro effectively addresses the multi-delay issue in two experiments.
> > >
> > > - Quantitative comparison in Tables 2 and 3. First, we perform an ablation analysis, as shown in Table 3. The results show that compared to scanning only in variable dimension, TimePro's MAE is reduced by 0.010. This result indicates that the adaptive time-tune strategy helps capture key time features and improves model performance. In particular, TimePro possesses a smaller number of parameters and GFLOPs than S-Mamba, which precludes performance improvement due to additional parameters and computational costs.
> > >
> > >   These quantified results suggest that the time-tune strategy can mitigate the multi-delay issue.
> > > - Qualitative comparison in Figure R7.
> > > We first add a visualization experiment to further validate the validity and interpretability of TimePro for the multi-delay issue, as shown in Figure R7 of the figure response. We choose test sequences from the ETTm1 and ETTh1 datasets as examples. Specially, we first calculate the correlation of label sequences (i.e., groundtruth) by Pearson correlation coefficient:
> > >
> > >   $r_{xy}=\frac{\sum_{i=1}^L(x_i-\overline{x})(y_i-\overline{y})}{\sqrt{\sum_{i=1}^L(x_i-\overline{x})^2\cdot\sum_{i=1}^L(y_i-\overline{y})^2}}$, where $x_i, y_i \in \mathbb{R}$ run through all time points of the paired variates to be correlated.
> > >
> > >      We then visualize the correlation between the variable features before and after HyperMamba. Figure R7 shows that TimePro selects important time points through the time-tune strategy, which drives the learned multivariate correlations closer to the label sequences. This suggests that TimePro effectively mitigates the detrimental effects of the multi-delay issue on accurate variable relationship modeling.
> > >
> > >   Besides, we also perform a visual ablation experiment in Fig. R6. It can be observed that TimePro's prediction curves are closer to the label curve than scanning only in the variable dimension or the time dimension.
> > >
> > >   Furthermore, TimePro's prediction curve is more similar to the ground truth in Fig. R4 and R5 . In contrast, S-Mamba and iTransformer map the variables as coarse embeddings, ignoring the delayed impact of different time points of each variable on the predicted sequence, leading to poorer results.
> > >
> > >   These qualitative results effectively validate TimePro's ability to mitigate the multi-delay problem.
> > >
> > > We will add detailed definitions and references of the multi-delay issue, Figure R7 and corresponding analyses to the final version. If the rebuttal helps address your concerns, we kindly ask that you increase your score to give TimePro a fighting chance!

---

### Official Review · Reviewer_Kihx · 2025-03-10

**Overall Recommendation:** 4

**Summary:**

This paper introduces TimePro, a novel Mamba-based framework designed for multivariate long-term time series forecasting. The core contribution lies in its variable- and time-aware hyper-state mechanism, which dynamically refines hidden states by adaptively selecting critical temporal intervals to address the multi-delay problem. Extensive experiments across popular benchmarks demonstrate superior performance over existing methods.

## update after rebuttal

I will keep my score.

**Claims And Evidence:**

The claims are supported by evidence.

**Essential References Not Discussed:**

N/A

**Experimental Designs Or Analyses:**

Reasonable experimental designs and analyses.

**Methods And Evaluation Criteria:**

The proposed method is evaluated on five widely used real datasets, including ETT (4 subsets), Exchange, Electricity, Weather, and Solar-Energy.

**Other Comments Or Suggestions:**

Figure 6 is insufficiently clear, making it difficult to discern critical details.

**Other Strengths And Weaknesses:**

**Strengths**

1.	The proposed method is technically sound.

2.	Extensive experiments across popular benchmarks demonstrate superior performance over existing methods.

3.	Detailed ablation studies justify the design choices.

**Weaknesses**

1.	I am not sure that the improvement over S-Mamba is significant enough, e.g., 0.251 vs. 0.250 for Weather and 0.398 vs. 0.392 for ETTm1.

2.	In Fig. 3“It consists of two parts, including plain state acquisition in the GPU SRAM and time tuning in the GPU HBM.” However, there are no details for this statement and it is not easy to understand.

**Questions For Authors:**

N/A

**Relation To Broader Scientific Literature:**

N/A

**Theoretical Claims:**

N/A

---

> ### Author Rebuttal · Authors · 2025-04-01
>
> We sincerely thank you for your time and efforts in reviewing our work and providing thoughtful feedback that can further strengthen our manuscript. We have added some experiments following your suggestions and they are available at  https://anonymous.4open.science/r/Anonymous_figure-2319/figure_response.pdf. Please see our detailed responses to your comments below.
>
> ### 1 Analysis of the improvement
> First, TimePro outperforms S-Mamba on seven datasets, and the improvement is significant on some datasets. For example, TimePro has a 0.009 and 0.015 improvement in MSE and MAE for the ETTh1 dataset, respectively. In addition, TimePro has a 0.011 and 0.008 improvement in MSE and MAE, for the Exchange dataset, respectively. These improvements fully validate the effectiveness of TimePro.
>
> Second, as mentioned in Fig. 5 of the manuscript, TimePro can significantly benefit from longer lookback window lengths because it preserves more local details in the time dimension. So, when the lookback window length is 96, TimePro's improvement on the Weather and ETTm1 datasets is 0.001 and 0.006, respectively. However, when the lookback window length is 336, the improvement is 0.008 and 0.01, respectively. This improvement is significant and further validates TimePro's potential. Finally, we provide a detailed efficiency analysis in Figure R1 in the figure response. It shows that TimePro has a better efficiency performance compared to S-Mamba. In summary, TimePro strikes a better balance between efficiency and performance than S-Mamba.
>
> ### 2 details for this statement
> As shown in Figure R3 of the figure response, we add some details in Fig. R3 and modify the corresponding caption. These details include: 1) textual explanations of GPU SRAM and HBM; 2) additions to the formulas in the grey boxes so that the reader can understand the initial state generation process; 3) some symbolic additions such as the state $h$ and hyperstate $h_o$, which correspond to Eq. 10-13 in the manuscript; 4) optimizations of some process components to enhance aesthetics and clarity; 5) textual additions to the captions, which further enhances the reader's understanding of hyper-scan. In addition, given that the final version has an extra page, we will add a more detailed description in Sec. 4.2. I hope these modifications can solve your confusion.
>
> ### 3 Fig. 6 Blurring
> I'm sorry that Figure 6 isn't clear enough. We have uploaded the image in PDF format as shown in Figure R4 of the figure response. In addition, we additionally upload the visualization comparison experiment on the ECL dataset in PDF format as shown in Figure R5. The clarity of these images is guaranteed. It can be observed that the prediction curve of TimePro is much closer to the groundtruth. This further validates the effectiveness of TimePro.
>
> We will add the above modifications and the corresponding analyses to the final version. If the rebuttal helps address your concerns, we kindly ask that you increase your score to give TimePro a fighting chance!

---

> > ### Comment · Reviewer_Kihx · 2025-04-06
> >
> > I am satisfied with the author's rebuttal, and decide to raise my score.

---

> > > ### Author Response · Authors · 2025-04-07
> > >
> > > # Thanks for your feedback
> > > Thank you very much for raising the rating. Your thoughtful comments have helped to improve this paper a lot!

---

### Official Review · Reviewer_FUHr · 2025-03-12

**Overall Recommendation:** 4

**Summary:**

This paper proposes TimePro, a Mamba-based model for multivariate long-term time series forecasting. By introducing a hyper-state mechanism that adaptively selects critical temporal intervals, TimePro aims to address the multi-delay problem, where variables influence targets over heterogeneous time spans. Empirical results across various benchmarks demonstrate competitive performance compared to Mamba- and Transformer-based baselines, with claims of linear computational complexity.

**Claims And Evidence:**

1. TimePro achieves state-of-the-art performance in multivariate long-term forecasting by adaptively modeling variable-specific temporal dependencies, which is justified by Table 2 and Fig. 1, where TimePro obtains superior MSE/MAE on various datasets.
2. This paper claims that the proposed hyper-state mechanism effectively captures both variable interactions and intra-variable temporal dynamics. Ablation studies (Tables 3–5) demonstrate that combining bidirectional variable scanning and time-aware offset learning reduces MSE and improves the performance.

**Essential References Not Discussed:**

NA.

**Experimental Designs Or Analyses:**

The experimental framework is rigorous, incorporating comprehensive ablation studies (e.g., feature dimensions, encoder depth) and complexity analyses to isolate the contributions of key components.

**Methods And Evaluation Criteria:**

This paper aligns with standard protocols (MSE/MAE metrics) across various benchmarks (e.g., ETT, Weather).

**Other Comments Or Suggestions:**

NA

**Other Strengths And Weaknesses:**

**Strengths**

1.	This paper is well written and the proposed method is technically sound.
2.	The integration of bidirectional variable scanning with time-aware offset learning effectively captures dynamic variable-temporal interactions, addressing a clear limitation in existing Mamba variants.
3.	Extensive experiments on diverse datasets (e.g., ETT, Weather) demonstrate the effectiveness of the proposed method. The design choices are well justified by ablation studies.

**Weaknesses**

1.	While theoretical complexity is analyzed (Table 1), real-world metrics (e.g., inference speed) are absent, which are important for justifying the advantages of the proposed method.
2.	The hardware-aware implementation of hyper-scan (Fig. 3) lacks details and poses challenges for understanding this part.
3.	As shown in Fig. 4, For the Exchange dataset, increasing the number of layers from 1 to 2 obtains slightly poorer performance. More discussions and insights are encouraged.

**Questions For Authors:**

NA

**Relation To Broader Scientific Literature:**

TimePro advances Mamba-based time series forecasting by addressing the multi-delay problem through dynamic time-aware hyper-states.

**Theoretical Claims:**

NA

---

> ### Author Rebuttal · Authors · 2025-04-01
>
> We sincerely thank you for your time and efforts in reviewing our work and providing thoughtful feedback that can further strengthen our manuscript. We have added some experiments following your suggestions and they are available at  https://anonymous.4open.science/r/Anonymous_figure-2319/figure_response.pdf. Please see our detailed responses to your comments below.
>
> ### 1 real-world metrics are absent
> Concerning your comments, we have added two experiments, including Figures R1 and R2 in the figure response.
> In Figure R1, we provide a detailed comparison of efficiency metrics including parameters, FLOPs, training time and inference time. It can be observed that TimePro possesses better computational efficiency than previous methods. Specifically, TimePro has a training time and inference time similar to S-Mamba and significantly outperforms other convolutional or transformer-based methods. For example, TimePro obtains lower prediction errors with 2.7 and 14.4 times the inference speed of PatchTST and TimesNet, respectively. Moreover, TimePro has the minimal parameters, FLOPs, and memory consumption. For example, TimePro requires only 67% parameters and 78% GFLOPs of S-Mamba. In addition to satisfactory efficiency, TimePro also outperforms recent advanced models including iTransformer and S-Mamba in forecasting performance. These results demonstrate TimePro's lightweight and suitability for deployment in a variety of real-world scenarios where resources are constrained.
>
> In Figure R2, we further explore the efficiency of TimePro with different variable channels, which can be seen as a complement to Table 1 in our manuscript. It can be observed that as the number of variable channels increases, iTransformer shows a quadratic increase in both the memory and inference time, which severely compromises the efficiency of the model and limits practical applications. In addition, PatchTST has unsatisfactory efficiency under all variable channels. In contrast, the linear scaling ability compared to variable channels, small memory consumption and high inference speed validate TimePro's efficiency.
>
> ### 2 lacks details in Fig. 3
> Thanks to your comments, we have added some details in Figure R3 in the figure response. These details include: 1) textual explanations of GPU SRAM and HBM; 2) additions to the formulas in the grey boxes so that the reader can understand the initial state generation process; 3) some mathematical symbol additions such as the state h and hyperstate ho, which correspond to Eq. 10-13 in the manuscript; 4) optimizations of some process components to enhance aesthetics and clarity; 5) textual additions to the captions, which further enhances the reader's understanding of hyper-scan.
>
> ### 3 More discussions of Fig. 4
> In Figure 4, as the number of layers increases from 1 to 4, the prediction error of the model first decreases and then gradually increases. This is due to the fact that when the number of layers is 1, the model is shallow and can not capture complex time-varying and variable relationships. When the number of layers is large, for example, 3 or 4, the model suffers from overfitting, which impairs the model's generalizability to the test set, and therefore a slight increase in prediction error occurs.
>
> We will add the above experiments and the corresponding analyses to the final version. Thanks again for your constructive suggestions!

---

### Official Review · Reviewer_gQxg · 2025-03-12

**Overall Recommendation:** 4

**Summary:**

This paper proposes TimePro, a Mamba-based model for multivariate long-term time series forecasting. TimePro adaptively selects critical time points to refine variable states, preserving temporal granularity and capturing dynamic variable relationships. Experiments on various benchmarks show competitive performance with existing Mamba- and Transformer-based methods.


## =================update after rebuttal==============

Thanks for the authors' responses. My concerns have been well addressed.

**Claims And Evidence:**

This paper claims TimePro achieves state-of-the-art results with linear complexity. Table 2 reports superior MSE/MAE across datasets. Complexity analysis (Table 1) confirms linear scaling with sequence length.

**Essential References Not Discussed:**

NA.

**Experimental Designs Or Analyses:**

The experimental designs and analyses are technically sound.

**Methods And Evaluation Criteria:**

This paper follows standard evaluation criteria with prior methods.

**Other Comments Or Suggestions:**

See weakness.

**Other Strengths And Weaknesses:**

**Strengths**

1.	This paper is well organized and well-written.

2.	The idea of incorporating variable and time-aware superstate construction is reasonable and interesting.

3.	Comprehensive ablation studies (e.g., feature dimensions, encoder layers) provide insights into design choices.

**Weaknesses**

1.	Limited discussion on computational overhead (e.g., training time, inference time) compared to baselines.

2.	Is it possible to provide some visualization results for better understand the proposed modules?

**Questions For Authors:**

NA.

**Relation To Broader Scientific Literature:**

The work builds effectively on Mamba-based models for multivariate long-term time series forecasting.

**Theoretical Claims:**

NA.

---

> ### Author Rebuttal · Authors · 2025-04-01
>
> We sincerely thank you for your time and efforts in reviewing our work and providing valuable feedback that can further strengthen our manuscript. We have added figures with more experiments following your suggestions and they are available at  https://anonymous.4open.science/r/Anonymous_figure-2319/figure_response.pdf. Please see our detailed responses to your comments below.
>
> ### 1 computational overhead discussion
> With reference to your comments, we have added two experiments, including Figures R1 and R2 in the figure response.
> In Figure R1, we provide a detailed comparison of efficiency metrics including parameters, FLOPs, training time and inference time. It can be observed that TimePro possesses better computational efficiency than previous methods. Specifically, TimePro has a training time and inference time similar to S-Mamba and significantly outperforms other convolutional or transformer-based methods. For example, TimePro obtains lower prediction errors with 2.7 and 14.4 times the inference speed of PatchTST and TimesNet, respectively. Moreover, TimePro has the minimal parameters, FLOPs, and memory consumption. For example, TimePro requires only 67% parameters and 78% GFLOPs of S-Mamba. In addition to satisfactory efficiency, TimePro also outperforms recent advanced models including iTransformer and S-Mamba in forecasting performance. These results demonstrate TimePro's lightweight and suitability for deployment in a variety of real-world scenarios where resources are constrained.
>
> In Figure R2, we further explore the efficiency of TimePro with different variable channels, which can be seen as a complement to Table 1 in our manuscript. It can be observed that as the number of variable channels increases, iTransformer shows a quadratic increase in both the memory and inference time, which severely compromises the efficiency of the model and limits practical applications. In addition, PatchTST has unsatisfactory efficiency under all variable channels. In contrast, the linear scaling ability compared to variable channels, small memory consumption and high inference speed validate TimePro's efficiency.
>
> ### 2 visualization effect of Hyper-Scan
> We provide some visualization results to validate the effectiveness of Hyper-Scan, as shown in Figure R6 in the figure response. It can be observed that suboptimal results are obtained for the prediction curves of models that scan only in the time dimension or the variable dimension. Specifically, when scanning only in the variable dimension (Fig. a), the model's prediction curves are smoother, with a poorer fit at the extremes. We attribute this to the model's lack of ability to capture the details of local changes within variables. When scanning in the time dimension only (Fig. b), the model's prediction ability for the extremes improves, but the average accuracy remains poor. This is due to the lack of capturing complex variable relationships. And when we use non-adaptive hyper-scanning (Fig. c), the model's average accuracy and predictive ability for extreme values are improved, but the performance is still unsatisfactory. And when we apply the adaptive hyper-scan, i.e., TimePro (Fig. d), the model can perceive both variable relationships and significant temporal information, resulting in a more accurate prediction performance.
>
> We will add these experiments (i.e., Figures R1, R2, R6) and the corresponding analyses to the final version. Thanks again for your valuable comments!

---

### Decision · Program_Chairs · 2025-05-01

**Decision:**

Accept (poster)

**Comment:**

This paper introduces TimePro, a Mamba-based model designed for forecasting multivariate long-term time series. It uses a hyper-state mechanism to adaptively choose important temporal intervals, targeting the multi-delay issue where variables affect targets across different time spans. Experiments on multiple benchmarks show that TimePro performs comparably to Mamba- and Transformer-based baseline models, and it is claimed to have linear computational complexity.

All reviewers agree to accept this work. One reviewer has raised his score (from weak reject to weak accept) and all problems have been solved.

Thus, the AC recommend‌s acceptance of this work.